Resource

# Differential assembly of mouse and human tumor microenvironments

Tristan Courau [1,2,3] ✉, Rebecca G. Jaszczak[3], Bushra Samad [2,3], Emily Flynn[3], Nayvin W. Chew[1,2,3], Gabriella C. Reeder[1,2,3], Jessica Tsui[1,2,3], Semhar Teklu[1,2], Lomax F. Pass[1,2], Austin W. Edwards[3], Mohammad Naser[3], Arja Ray[1,2], Harrison Wismer[1,2,3], Daniel Bunis[3], Leonard Lupin-Jimenez[3], Noah V. Gavil[4], David Masopust [4], John P. Graham[5], Daniel A. Skelly [5], Xavier Vesco [5], Edison T. Liu [6], Gabriela K. Fragiadakis [3,7,9], Alexis J. Combes [1,2,3,8,9] & Matthew F. Krummel [1,2,9] ✉

Mouse models are frequently used to develop treatments for human cancer. However, the degree to which their tumor microenvironments (TMEs) are synonymously assembled is particularly poorly characterized. Through systematic immunoprofiling of 15 commonly used mouse models, we found that most murine TMEs recapitulate the composition of poorly infiltrated human tumors, extensively biased toward high macrophage densities. We discovered substantial species-specific biases of chemokine expression networks known to drive TMEs assembly, together with discoordinated frequencies of T and myeloid cell subtypes. Even with variable alignment, conserved cell-type-specific gene expression programs emerged across species and cohorts. Dissecting the coordinated T cell–myeloid gene expression programs revealed a conserved axis between interferon-responsive myeloid states and ongoing T cell cytotoxicity that transcends tissue of origin and predicts clinical outcome. Collectively, this work provides a practical atlas outlining both the hazards and opportunities of using mice to model human cancer.

The immune composition of the human tumor microenvironment (TME) is a critical determinant of patient outcomes[1,2]. Human TMEs span a spectrum from highly inflamed, immune-rich ('hot') tumors to multiple poorly infiltrated ('cold') types, further distinguished by fibrotic components, and T cell and myeloid subpopulations[3–7]. These classes associate with prognosis across cancer types[3–7], and each will probably require tailored therapeutic consideration. However, the mechanisms driving these recurring TME patterns remain poorly understood, limiting feature-based treatment design[8,9].

Despite millions of years of evolutionary divergence, mice remain indispensable for studying human cancer biology. Mouse models have revealed conserved mechanisms of tumor development, progression and drug resistance[10], in addition to demonstrating the efficacy of checkpoint blockade and fundamentals of the tumor immune escape[11–15]. Yet, frequent failures in predictive efficacy[16,17] highlight the need for more nuanced model selection and deeper understanding of species-specific differences in tumor immunity.

In-depth comparison between murine and human TMEs have been limited. Prior studies report immune diversity across mouse models and tumor sizes[18–22], as well as discrepancies in T cell profiles with indication-matched patients[23]. While insightful, they lack sufficient

[1]Department of Pathology and ImmunoX Initiative, UCSF, San Francisco, CA, USA. [2]ImmunoProfiler Initiative, UCSF, San Francisco, CA, USA. [3]CoLabs, UCSF, San Francisco, CA, USA. [4]Department of Microbiology and Immunology, University of Minnesota Medical School, Minneapolis, MN, USA. [5]The Jackson Laboratory for Mammalian Genetics, Bar Harbor, ME, USA. [6]The Jackson Laboratory for Genomic Medicine, Farmington, CT, USA. [7]Department of Medicine, Division of Rheumatology, UCSF, San Francisco, CA, USA. [8]Department of Medicine, Division of Gastroenterology, UCSF, San Francisco, CA, USA. [9]These authors jointly supervised this work: Gabriela K. Fragiadakis, Alexis J. Combes, Matthew F. Krummel. ✉e-mail: tristan.courau@ucsf.edu; max.krummel@ucsf.edu

breadth and resolution to assess how well mouse models capture the full diversity of human TMEs.

Several frameworks classify human TMEs into conserved 'subtypes'[3,4], 'ecotypes'[6] or 'archetypes'[7], which presumably represent templates for local immune systems that co-opt immune contexts ranging from wound healing to chronic viral infections[9,24–29]. These provide considerably greater granularity than the broad immune 'rich' versus 'poor' designations. Another approach to describe TME diversity is characterizing their defining cell–cell relationships. For example, regulatory T ($T_{reg}$) and exhausted T cells have each been linked to particular myeloid populations in some human and murine TMEs[30–33], while in others the occurrence of tertiary lymphoid structures is mediated through co-inclusion of dendritic cells, various fibroblasts populations and CD4 T cells expressing the B-cell-attracting chemokine CXCL13 (refs. [34–37]).

To systematically compare TMEs across species, we generated high-dimensional profiles from diverse murine TMEs and analyzed them alongside human TME datasets. By integrating archetype classification, gene expression programs (GEPs) and cell–cell relationships, we identified consistent species biases, including differences in chemokine expression (notably *CXCL13*), immune interactions and transcriptional programs. Altogether, this resource provides both rich data (accessible at https://quipi.org/app/quipi_humu) and a framework for understanding similarities and divergences between mouse and human TMEs, supporting more informed selection and refinement of mouse models for translational cancer research.

## Results

### Most mouse models recapitulate the composition of desertic, macrophage-rich human tumors

To benchmark the immune composition and transcriptomic patterns of typical murine TMEs against human TMEs, we profiled 15 widely used mouse models representing more than 95% of published immunotherapy studies. This selection includes common cell lines (notably B16F10, MC38, CT26, LLC, 4T1 or RENCA), autochthonous/transplanted (KPC) and genetically engineered models (MMTV-PyMT) across BALB/c and C57BL/6 backgrounds[18–20,23,38] (Fig. 1a, Table 1 and Supplementary Table 1). Tumors were analyzed at day 14 after implantation (or ~500 mm³) using cytometry by time-of-flight (CyTOF) in all models and using single-cell RNA sequencing (scRNA-seq) in nine models.

As a primary comparator, we used the human ImmunoProfiler dataset, which defines immune archetypes based on human TME composition and deep transcriptomic profiling of T cell, non-granulocytic myeloid, stromal and tumor populations, and has already benchmarked multiple studies[7,31,39,40] (Supplementary Table 1). By analyzing total immune frequencies across human and murine TMEs (Extended Data Fig. 1a), we found that both species displayed bimodal distributions of total immune density ('rich' and 'poor'), but murine TMEs had significantly lower overall immune cell frequencies than human TMEs (Fig. 1b).

Based on our studies and others that classified human TMEs[4,6,7], we used ten major cellular TME compartments to compare the composition of overall human and murine TMEs (Extended Data Fig. 1b). We found that murine TMEs consistently contained fewer T cells and higher myeloid frequencies than human TMEs, with a particular bias toward macrophage abundance (Fig. 1c). Variance across murine TMEs was driven largely by myeloid density and composition, whereas human TMEs exhibited broader T cell variability (Fig. 1d). Imaging of intact human and murine tumors confirmed these findings, showing that human tumors generally were T-cell-dominated, while mouse tumors were strongly myeloid-biased (Fig. 1e,f and Extended Data Fig. 1j).

Despite model-specific variations in murine TME composition (Extended Data Fig. 1c,d, in line with previous reports[18–20,23]) embedding mouse and human samples revealed that nearly all mouse models (except RENCA) clustered with immune-desert, macrophage-rich human archetypes (Fig. 2a,b). These two archetypes represent only ~17% of patients in our ImmunoProfiler cohort[7], with indication-specific variations ranging from 0% in hepatocellular carcinomas to 46.5% in gynecological tumors (Extended Data Fig. 1e).

To extend this analysis, we queried 2,543 additional mouse RNA-seq samples from 178 studies deposited in the NCBI's Sequence Read Archive that we benchmarked against an established human TME classification[3] built on the TCGA[41]. Across this large mouse cohort, more than 60% were classified as 'lymphocyte-depleted' (C4), while very few were classified as 'immune-rich' (~7% in the C2 and C6 subtypes) (Fig. 2c, Extended Data Fig. 1h,i and Supplementary Table 1). Therefore, low T cell infiltration is a dominant feature of murine tumors.

Importantly, the T cell to myeloid imbalance in murine TME (Extended Data Fig. 1f) persisted across many experimental conditions, including KPC pancreatic tumors implanted either orthotopically or subcutaneously, and in B16, LLC and MC38 tumors inoculated to either mice housed under 'dirty' conditions, which can promote T cell activation and accumulation in tissues[42], or to aged mice or mice fed a high-fat diet (Extended Data Fig. 1g).

Taken together, these findings suggest that even though murine TMEs display model-specific variations, they predominantly resemble a minority subset of immune-desert, macrophage-rich human TMEs.

### Divergence in chemokine networks

Chemokine networks being major contributors to immune cell densities in tissues[43], we performed a systematic analysis of chemokine and receptor transcript expression[7,43,44] within key cell populations of the TMEs. For this, we analyzed bulk RNA-seq data from sorted T, $T_{reg}$, myeloid, tumor and stroma compartments from human TMEs[7], compared to pseudobulked scRNA-seq datasets from our murine TME cohort (Extended Data Fig. 2a). The mouse scRNA-seq also allowed to describe subtype-specific expression of each chemokine (Extended Data Figs. 2b–f and 3b–d) in T cells, dendritic cells, monocytes and macrophages, and nonimmune cells subclusters.

We first analyzed the expression patterns of chemokine receptors across compartment and species (Extended Data Fig. 3a–c), before focusing on T-cell-specific chemokine receptors. While several chemokine networks were conserved (*CCR4*, *CCR7*, *CXCR4*, *CXCR6* expression patterns, as well as their ligands), key differences emerged. Notably, *CCR2* and *CCR5* transcripts were reduced in mouse T cells relative to human T cells (Fig. 3a,b), which was consistent across models and subsets (Extended Data Fig. 3b) and confirmed at the protein level for *CCR5* (Fig. 3c).

In addition, the expression of *CXCR3* and *CXCR5* (two major drivers of T cell infiltration in tumors[43–46]) was conserved in murine T cells but the expression patterns of their canonical ligands were highly dissimilar across species. In murine TMEs, *CXCL9* and *CXCL10*, and *CXCL13*, were enriched in the stroma (that is, fibroblasts), while biased toward myeloid and T/$T_{reg}$ cells, respectively, in human TMEs (Fig. 3d,e and Extended Data Fig. 3c,d). The T cell bias for *CXCL13* expression in human TMEs was not absolute because occasional human tumors showed substantial stromal *CXCL13* expression (Fig. 3f). These observations bear an immediate importance for the field as recent studies suggested that immune checkpoint responsive networks are organized around CXCL13⁺ T cells in patients[47–51] and those rarely occur in mouse models. They also constitute important guidance for future mouse studies of human-relevant transcriptomic networks.

### Interspecies deviations in TME immune cellular networks

Multiple studies have demonstrated that the presence of one cell type in the TME can correlate with another, typically because one either recruits or supports the other[31,52,53]. Therefore, we analyzed specific relationships in cell densities between cell types in human versus murine TMEs (Fig. 4). Total immune infiltration was inversely

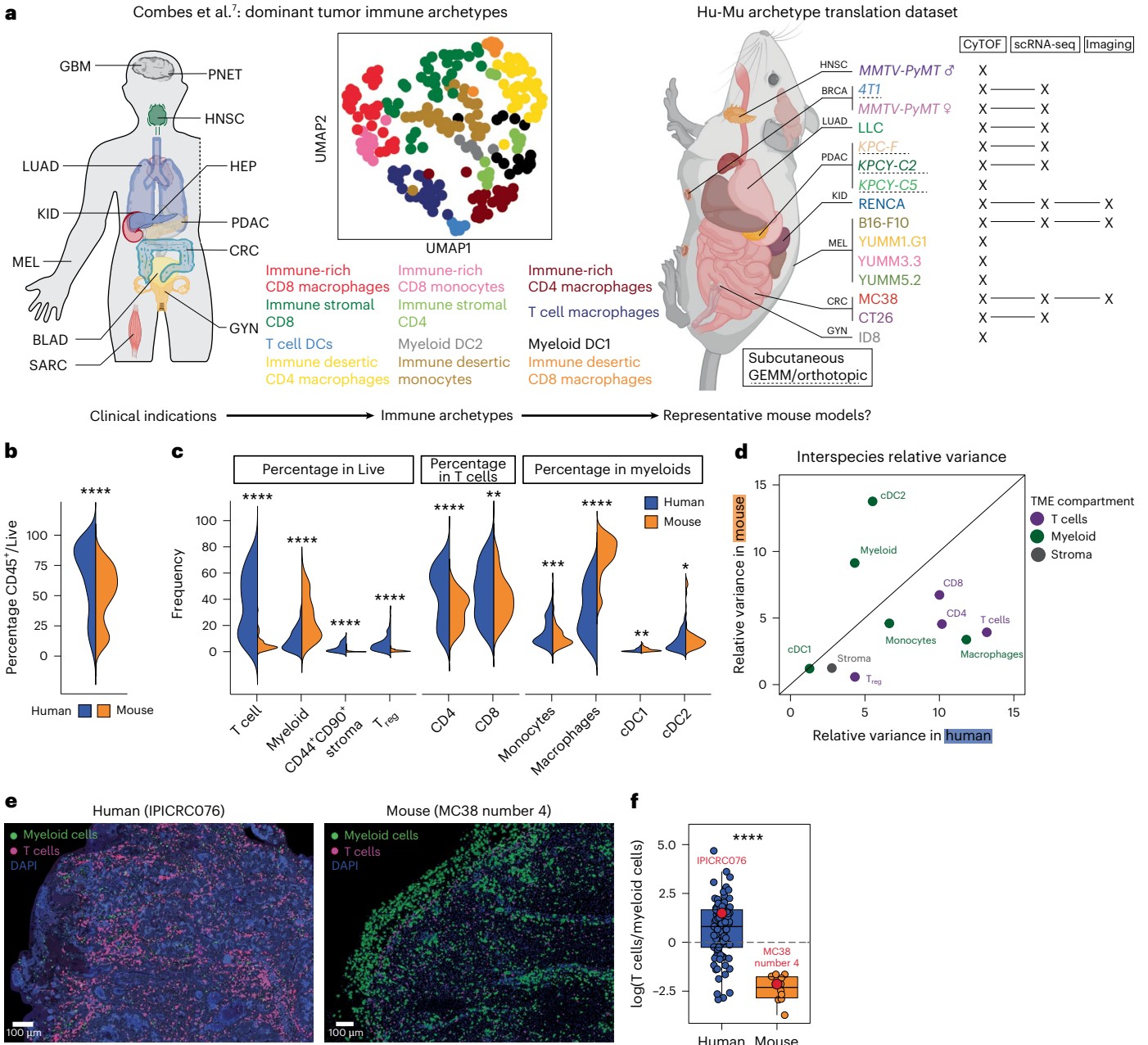

**Fig. 1 | Compositional disparities between human and murine tumors.**
**a**, Schematic of our human[7] and murine study cohorts. **b**, Violin plot presenting the frequency of CD45[+] in Live cells from human (blue, all samples grouped, $n = 170$) versus murine (orange, all samples grouped, $n = 109$) tumors ($P = 1.39 \times 10^{-7}$). **c**, Violin plot presenting the frequencies of conventional T cells ($P = 4.83 \times 10^{-55}$), non-granulocytic myeloid cells (combining monocytes, macrophages, conventional DCs and plasmacytoid DCs, $P = 2.57 \times 10^{-19}$), nonimmune stroma (CD44[+]CD90[+] in CD45[-], $P = 2.17 \times 10^{-30}$) and CD4[+] T_reg cells ($P = 1.65 \times 10^{-35}$) out of Live cells; CD4 T conventional ($P = 4.23 \times 10^{-8}$) and CD8 ($P = 7.31 \times 10^{-4}$) out of T cells; monocytes ($P = 9.55 \times 10^{-4}$), macrophages ($P = 1.72 \times 10^{-47}$), cDC1 ($P = 5.85 \times 10^{-3}$) and cDC2 ($P = 4.11 \times 10^{-2}$) out of myeloid cells between human (blue, all samples grouped, $n = 224$) and mouse (orange, all samples grouped, $n = 109$) tumors. **d**, Plot comparing the relative variance of each parameter shown in **b** (colored according to cellular compartment) between human and murine tumors. The diagonal line represents an equal relative variance between the two species. **e**, Representative images showing our identification of T (pink, identified as CD3[+]CD4[+/−]CD8[+/−] in human and CD3[+]CD11b[−] in mouse) and myeloid (green, identified as

CD3[−]HLA-DR[+]XCR1[+/−]CD163[+/−] in human and CD3[−]CD11b[+] or CD3[−]MHC-II[+] in mice) cells in a human (left) or murine (right) tumor slice. **f**, Box plot presenting the ratio of T cells over myeloid cells in human (blue, $n = 85$) versus murine (orange, $n = 10$) tumors ($P = 2.1 \times 10^{-6}$), calculated using a measurement based on the imaging shown in **e**. Boxes represent the 25th–75th percentile; the horizontal line represents the median; the whiskers represent 1.5 times the interquartile range (Tukey); and points represent individual samples. Statistical significance in all panels was calculated using a *t*-test with Bonferroni correction, $^*P_{adj} \le 0.05$, $^{**}P_{adj} \le 0.01$, $^{***}P_{adj} \le 0.001$, $^{****}P_{adj} \le 0.0001$. All statistical tests were two-sided; *P* values are reported as exact values unless otherwise indicated. GBM, glioblastoma; PNET, primitive neuroectodermal tumor; HNSC, head and neck squamous cell carcinoma; LUAD, lung adenocarcinoma; HEP, hepatic tumor; KID, kidney tumor; PDAC, pancreatic ductal adenocarcinoma; MEL, melanoma; CRC, colorectal cancer; BLAD, bladder cancer; GYN, gynecologic tumor; SARC, sarcoma; UMAP, uniform manifold approximation and projection; Hu, human; Mu, murine; BRCA, breast carcinoma; GEMM, genetically engineered mouse model; cDC, conventional dendritic cells. Panel **a** created in BioRender; Lab, C. https://biorender.com/vf8ffoa (2026).

**Table 1 | Cell lines**

| Cell line | Origin | Culture medium | Incubation | Injection site | Number injected |
|---|---|---|---|---|---|
| B16-F10 | CRL-6475 (ATCC) | DMEM (catalog number 11995-065, Gibco), 10% FCS (Benchmark), 1× penicillin-streptomycin-glutamine (Thermo Fisher Scientific) | 37 °C 5% $CO_2$ | Subcutaneous, right flank | 250,000 |
| LLC | CRL-1642 (ATCC) | | 37 °C 5% $CO_2$ | Subcutaneous, right flank | 500,000 |
| MC38 | SCC172 (Sigma-Aldrich) | | 37 °C 5% $CO_2$ | Subcutaneous, right flank | 500,000 |
| 4T1 | CRL-2539 (ATCC) | | 37 °C 5% $CO_2$ | Mammary fat pad | 250,000 |
| YUMM1.G1 | CRL-3363 (ATCC) | | 37 °C 5% $CO_2$ | Subcutaneous, right flank | 2 million |
| YUMM3.3 | CRL-3365 (ATCC) | | 37 °C 5% $CO_2$ | Subcutaneous, right flank | 2 million |
| YUMM5.2 | CRL-3367 (ATCC) | | 37 °C 5% $CO_2$ | Subcutaneous, right flank | 2 million |
| KPC-F (FC1245) | E. Collisson (UCSF) | | 37 °C 5% $CO_2$ | Pancreas (orthotopic injections described in Jiang et al.[38]) | 1,000 |
| KPCY-C2 (6694c2) | | | 37 °C 5% $CO_2$ | | 500,000 |
| KPCY-C5 (7160c5) | | | 37 °C 5% $CO_2$ | | 500,000 |
| ID8 | SCC145 (Thermo Fisher Scientific) | DMEM, 4% FCS, 1× penicillin-streptomycin-glutamine, 1× insulin-transferrin-selenium (Thermo Fisher Scientific) | 37 °C 5% $CO_2$ | Subcutaneous, right flank | 2 million |
| RENCA | CRL-2947 (ATCC) | RPMI (catalog number 11875-093, Gibco), 10% FCS, 1× penicillin-streptomycin-glutamine | 37 °C 5% $CO_2$ | Subcutaneous, right flank | 500,000 |
| CT26 | CRL-2638 (ATCC) | | 37 °C 5% $CO_2$ | Subcutaneous, right flank | 500,000 |

correlated with proliferating tumor cells (measured using Ki-67 staining; Extended Data Fig. 4a) in mice and human TMEs (Fig. 4a,b). Similarly, both species exhibited positive correlations between CD8 T cell frequencies and their degree of exhaustion (Fig. 4c and Extended Data Fig. 4b), and a relationship between cDC1 and cDC2 frequencies versus CD8 and CD4 T cell frequencies, indicative of co-maturation of these cells in tumors as described previously[30,54] (Extended Data Fig. 4c).

However, several correlations differed between human and murine TMEs. For example, CD4 conventional T ($T_{conv}$) cell and $T_{reg}$ cell frequencies correlated in murine TMEs but not broadly in human TMEs (Extended Data Fig. 4d). More importantly, correlations between T cells and myeloid cells (Extended Data Fig. 4e) or between CD8 T cells and macrophages (Fig. 4d) in murine TMEs[31] were not global features of human TMEs. Although this could partly originate from differential myeloid cell identification in human TMEs versus murine TMEs (human myeloid cells are sorted as HLA-DR[+7] but not mouse myeloid cells), when restricting human samples to mouse-like immune-desert archetypes, the correlations were restored (insets in Fig. 4b–d and Extended Data Fig. 4c,e). This may indicate that these cells track one another but only under specific conditions, that is, in the absence of a large T cell pool or a global bias of the TME toward myeloid cells. Thus, these observations provide additional strong guardrails for interpreting murine tumor efficacy data for drug treatments that target these cell populations and their partners.

### Consensus nonnegative matrix factorization identifies robust, cross-species transcriptomic programs

To compare human TMEs versus murine TMEs at a more granular and unbiased level, we applied a consensus nonnegative matrix factorization (cNMF) analytical pipeline[5,55–57], anchoring on T cells and non-granulocytic myeloid cells to define their particular sets of GEPs and benchmark them cross-species (Fig. 5a).

When analyzing human T cells ($T_{reg}$ cells excluded[7]), we found nine stable GEPs (Extended Data Fig. 5a and Supplementary Table 2). We integrated Gene Ontology information with known genes functions to annotate these GEPs, such as human GEP T_3 was linked to 'T cell cytotoxicity' (*PRF1*, *LAG3*, *GZMB* and *NKG7*), while T_9 is linked to 'CD4 regulation' (*TSC22D3*, *JUNB*, *RGS1*, *IL7R*, *CD69*) (Extended Data Fig. 5c). The enrichment of some of these human GEPs correlated with TME

composition (Extended Data Fig. 6a). For example, T_3 'T cell cytotoxicity' and T_5 'CD4 T cell-associated' correlated profoundly with CD8 and CD4 T cell frequencies, respectively. 'CD4 regulation' correlated with stroma and $T_{reg}$ enrichment in the tumor, possibly representing inhibitory circuits in TMEs. Repeating this analysis in murine TMEs uncovered 25 stable T cell GEPs (Extended Data Fig. 5e), similarly associated with T cell function and TME composition (Extended Data Figs. 5g and 6d). To quantify cross-species overlap of these GEPs, we used a Jaccard similarity index and found high degrees of similarity in three of the nine human T cell GEPs (Fig. 5b, Extended Data Fig. 6g,i and Supplementary Table 2). This notably included T_3 'T cell cytotoxicity' (Fig. 5c) and T_9 'CD4 regulation' (Fig. 5d).

Applying the same approach to the myeloid compartment in human TMEs identified 14 stable GEPs (Extended Data Fig. 5b). These included My_1 'inflammatory' (*IL1A*, *IL1B* and *NLRP3*) and My_2 'IFN response' (*IFIT2*, *IFIT3*, *ISG15* and *CXCL10*) (Extended Data Fig. 5d and Supplementary Table 2). Some were highly correlated with cellular composition, including association of tumor-associated macrophage (TAM) densities with My_8 'lipid metabolism' and monocyte densities with My_6 ('migration') and My_9 ('regulation of defense response') (Extended Data Fig. 6b). In murine TMEs, we identified 23 distinct and stable GEPs (Extended Data Fig. 5f), again often driven by genes known for their association with myeloid functions or whose enrichment correlated with TME composition (Extended Data Figs. 5h–j and 6e). Akin to T cells, a Jaccard analysis demonstrated high degrees of similarity for four pairs of myeloid GEPs (Fig. 5e and Extended Data Fig. 6h), including My_2 'IFN response' (Fig. 5f) and My_1 'inflammatory' (Fig. 5g), and the TAM-associated My_8 'lipid metabolism' and My_11 'LYVE1 TAMs' (Extended Data Fig. 6j,k).

To assess the robustness of the conserved T cells/myeloid GEPs, we used a Jaccard analysis with an independent dataset describing various GEPs (or 'meta-programs') across populations, studies and tumor indications in human[58]. All our cross-species conserved human GEPs (that is, My_1, My_2, My_8, My_11, T_3, T_4 and T_9) found an equivalent in this dataset (Fig. 5h,i). This was also the case when comparing these GEPs to another recent study describing T cells and myeloid GEPs in patients with glioma[59] (Extended Data Fig. 7a), although sometimes our GEPs split into two in these other datasets, or conversely.

As the different platforms used to generate human and murine GEPs (bulk versus scRNA-seq, respectively) could cause the low degree

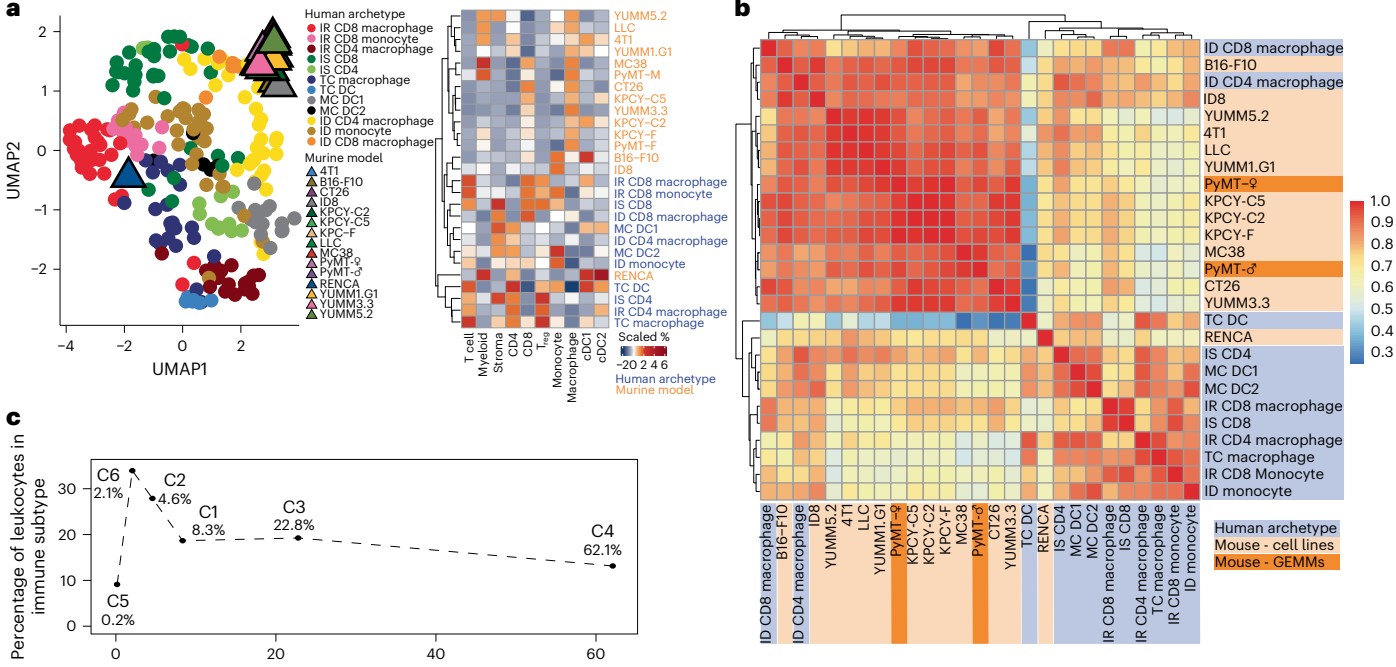

**Fig. 2 | Mapping mouse tumor models to human TME archetypes and subtypes. a**, Embeddings of the 10 feature frequencies in Fig. 1c showing human samples (circles, individual samples colored according to archetype, n = 224) and mouse tumors (triangles, initial n = 109 averaged according to tumor line resulting in n = 12 data points on the graph) in the same UMAP space (left) or in a hierarchically clustered heatmap (right). For both graphs, the 10 feature frequencies were z-scored across the entire dataset (224 human + 109 mouse samples) before averaging by human archetype/mouse model and UMAP

embedding/plotting. **b**, Hierarchically clustered matrix of cosine similarities calculated between each mouse tumor model (orange) and each human archetype (blue), using the frequencies in Fig. 1c. **c**, Scatter plot presenting the mapping of 2,543 mouse tumor samples from the NCBI Sequence Read Archive in the 6 immune subtypes described in ref. 3. The graph shows the percentage of mouse samples mapping to each subtype (x axis, values also displayed on the graph) and the average percentage of leukocytes infiltrated in each subtype (y axis).

of overlap seen in Fig. 5b,e, we used the recent 'meta-programs' study[58] to compare scRNA-seq-generated GEPs in mouse versus human (Extended Data Fig. 7b). This both validated the conservation of GEPs described above and found additional new similarities, notably in T cells, by linking murine T_16, T_19, T_15 and T_23, respectively to 'proliferative', 'naive', 'IFN-driven' and 'glycolytic'/'dysfunctional' human T cell GEPs. In myeloid cells this also highlighted My_14/My_19 as similar to myeloid proliferative GEPs[58]. To this extent, GEP analysis may prove a substantially improved way to categorize cell functions both across studies and across species.

Leveraging the high resolution of our murine scRNA-seq datasets, we described the usage of murine GEPs by each murine cell subtype described in Extended Data Fig. 2. This underlined varying patterns of GEP usage across cell populations (Extended Data Fig. 7c), including some GEPs that are extremely specific to a given subtype (for example, Mu_T_25 in γδ T cells) while other GEPs seem to be used by many different cell subsets (for example, Mu_T_9 across all CD4 T cells).

We then asked whether the robust, conserved GEPs found across studies and platforms could also be observed in other biological processes than tumors. Thus, we took advantage of a recent description of GEPs occurring over time and space in mouse myeloid cells during wound healing[24] and again applied the Jaccard analysis approach (Extended Data Fig. 7d). This showed that three of these— My_1 'inflammatory', My_2 'inflammatory' and My_11 'LYVE1 TAMs'—had a similar GEP in the process of wound healing in mice.

This set of analyses overall defines a robust collection of core-conserved GEPs that recur from mouse to human across studies and biological processes. We hypothesize that these may serve as modern measures of immune identity, in some cases providing metrics for describing overall immune status and similarity, with cross-species

conservation providing a tractable way to perturb and study the process in a model system.

**Intercellular GEP 'movements' and their clinical relevance**

We next examined coordinated correlations between T cell and myeloid GEPs[24], which we called 'movements', as exemplar and because of the known importance of this axis for antitumor immune responses[1,30,52,54]. Several such 'movements' were present in human TMEs (Fig. 6a and Extended Data Fig. 6c–f), but only one was clearly conserved in murine TMEs: the association between T_3 'T cell cytotoxicity' and My_2 'IFN response'. This conserved axis was particularly evident in IR CD8 mac and IS CD8 human archetypes and in MC38, CT26 and B16-F10 mouse models (Fig. 6b).

To assess the relevance of this GEP 'movement', we categorized patients as high or low for both T_3 'T cell cytotoxicity' and My_2 'IFN response' GEPs (Fig. 6c) and observed how their combinations parsed out overall survival. This showed a trend for improved survival of patients who displayed an overall high enrichment for both GEPs (T_3^Hi My_2^Hi) compared to any other combinations of these GEPs (Fig. 6d). Analyzing an independent transcriptomic dataset of whole human TMEs (TCGA[7,41]) demonstrated that these gene programs are also strongly correlated in this larger dataset (Fig. 6e). In this cohort, high T_3 'T cell cytotoxicity' was generally associated with better outcomes compared to lower, but the T_3^Hi My_2^Hi condition is the most favorable for outcome compared to all the other combinations (Fig. 6f). As described elsewhere[60,61], we found the relationship between cytotoxic T cells and IFN-stimulated myeloid cells to be beneficial for survival of patients with cancer.

Other human GEP correlations, such as T_9 'CD4 regulation' with My_1 'inflammatory', were not globally conserved in mice (Fig. 6a and

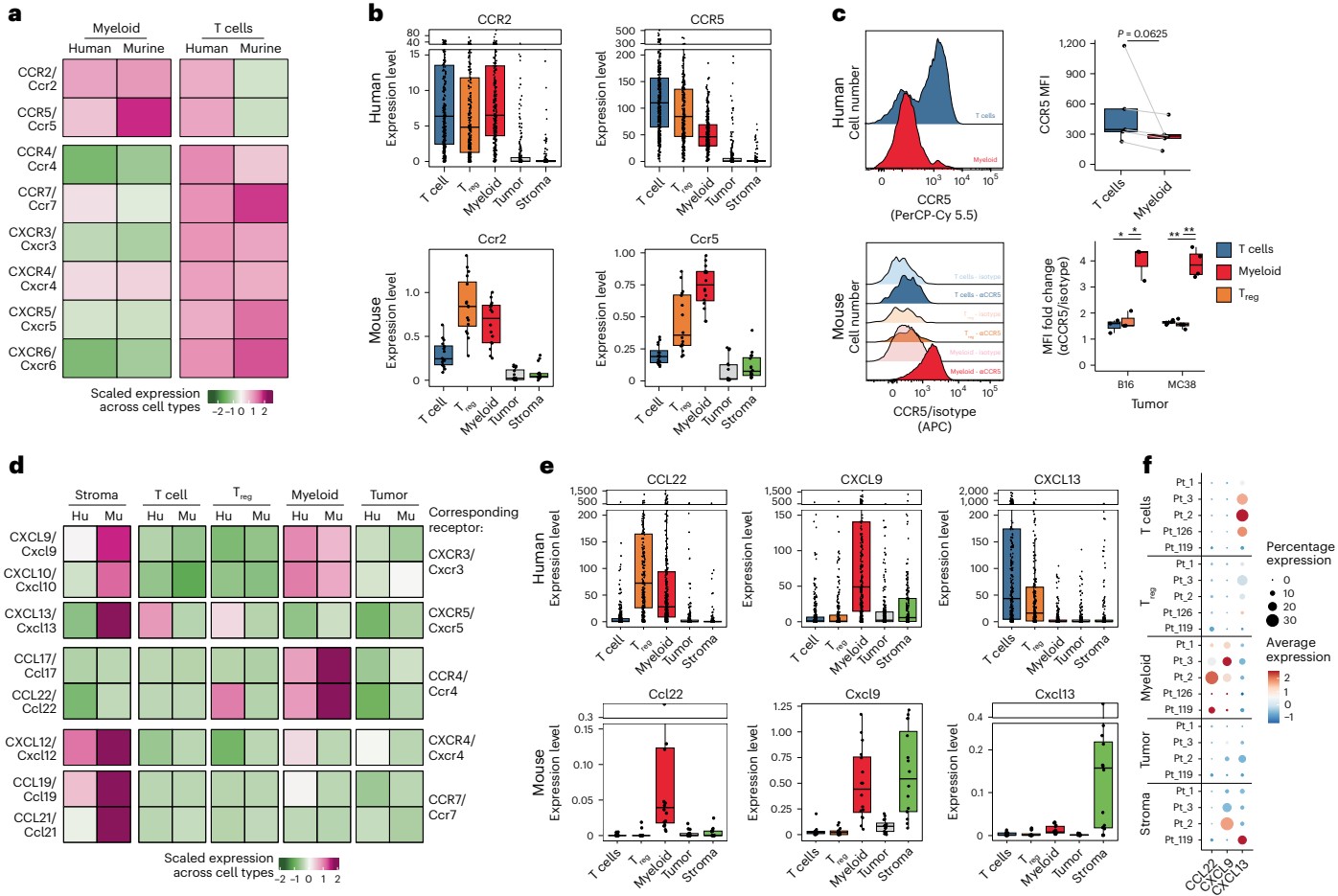

**Fig. 3 | Divergent chemokine networks underlie dysregulated T cell abundances in the TME. a**, Heatmap comparing the scaled expression of chemokine receptors between human and murine T cells. For each species, the expression of each receptor was extracted for all T cells (bulk-sorted in human versus pseudobulked from scRNA-seq in mice) and scaled compared with their expression in the other available TME compartments of the same species (that is, $T_{reg}$, myeloid, tumor and stroma). We then extracted the scaled values for T cells and plotted them side-by-side in a heatmap. The full heatmap can be found in Extended Data Fig. 3. **b**, Box plots showing the expression levels of CCR2 and CCR5 across T cells, $T_{reg}$, myeloid, tumor and stroma in human (top, shown as transcripts per million (TPM) from bulk RNA-seq, each dot representing a single patient) versus murine (bottom, shown as the expression level from scRNA-seq averaged per sample, each dot representing a single mouse sample) tumors, exemplifying the differences found in **a. c**, Flow cytometry validation of the protein expression of *CCR5* in human (top) and murine (bottom) tumors across different cell compartments, plotted as representative histograms (left)

and box plots of mean fluorescence intensity (MFI) (right). For the mouse box plot, T cells versus myeloid comparisons yielded $P = 0.013$ in B16 and $P = 0.0033$ in MC38; $T_{reg}$ versus myeloid comparisons yielded $P = 0.012$ in B16 and $P = 0.002$ in MC38. **d**, Heatmap (as in **a**) comparing the expression of specific chemokine ligands binding the receptors found conserved in **a** between the different cellular compartments of human versus murine tumors. The full heatmap can be found in Extended Data Fig. 3. **e**, Box plots as in **b** showing the expression levels of CCL22, CXCL9 and CXCL13 across cellular compartments and species. **f**, Dot plot presenting the scaled expression of CCL22, CXCL9 and CXCL13 across the cellular compartment of three different patients bearing HNSC tumors analyzed using scRNA-seq. For all box plot, boxes represent the 25th–75th percentile; the horizontal line represents the median; the whiskers represent 1.5 times the interquartile range (Tukey); and points represent individual samples. In **c**, statistical significance was calculated using a *t*-test with Bonferroni correction, *$P_{adj} \leq 0.05$, **$P_{adj} \leq 0.01$. All statistical tests were two-sided; *P* values are reported as exact values unless otherwise indicated.

Extended Data Fig. 7e,f). We again noted that when we censored human samples to include only those most similar to murine TMEs (ID CD4/CD8 mac), the correlation was less pronounced (Extended Data Fig. 7e). Interestingly, in the TCGA we found that patients with T_9^Hi^/My_1^Lo^ gene expression levels tended to have improved survival compared to other conditions (Extended Data Fig. 7g,h). CD4 T cell activation in the absence of concurrent IL-1-related myeloid inflammation could thus be optimal, which is consistent with previous reports associating poor outcome and inflammatory myeloid cells[62,63]. We also noted that the mouse-like archetypes (generally T_9^Lo^) were among the worst surviving, hinting that mouse models may be relevant to study these classes of poor survivors in patients.

Altogether, these analyses showcase the use of GEPs and their correlations as 'movements' as entry points to studying cellular

crosstalk in murine TMEs that is relevant to human TMEs and linked with patient outcomes.

## Discussion

This study produced a data resource and systemic analyses examining the degree and possible circumstances to which murine TMEs can represent human TMEs. We identified specific cellular and molecular shortcomings of mouse models that were not previously well defined, while also highlighting conserved immune relationships and gene programs that align with specific human TMEs and can be studied to understand principles related to therapeutic strategies. Together, our resource provides guidance for selecting and refining models based on the human biology they best replicate. Users may independently query this data at https://quipi.org/app/quipi_humu.

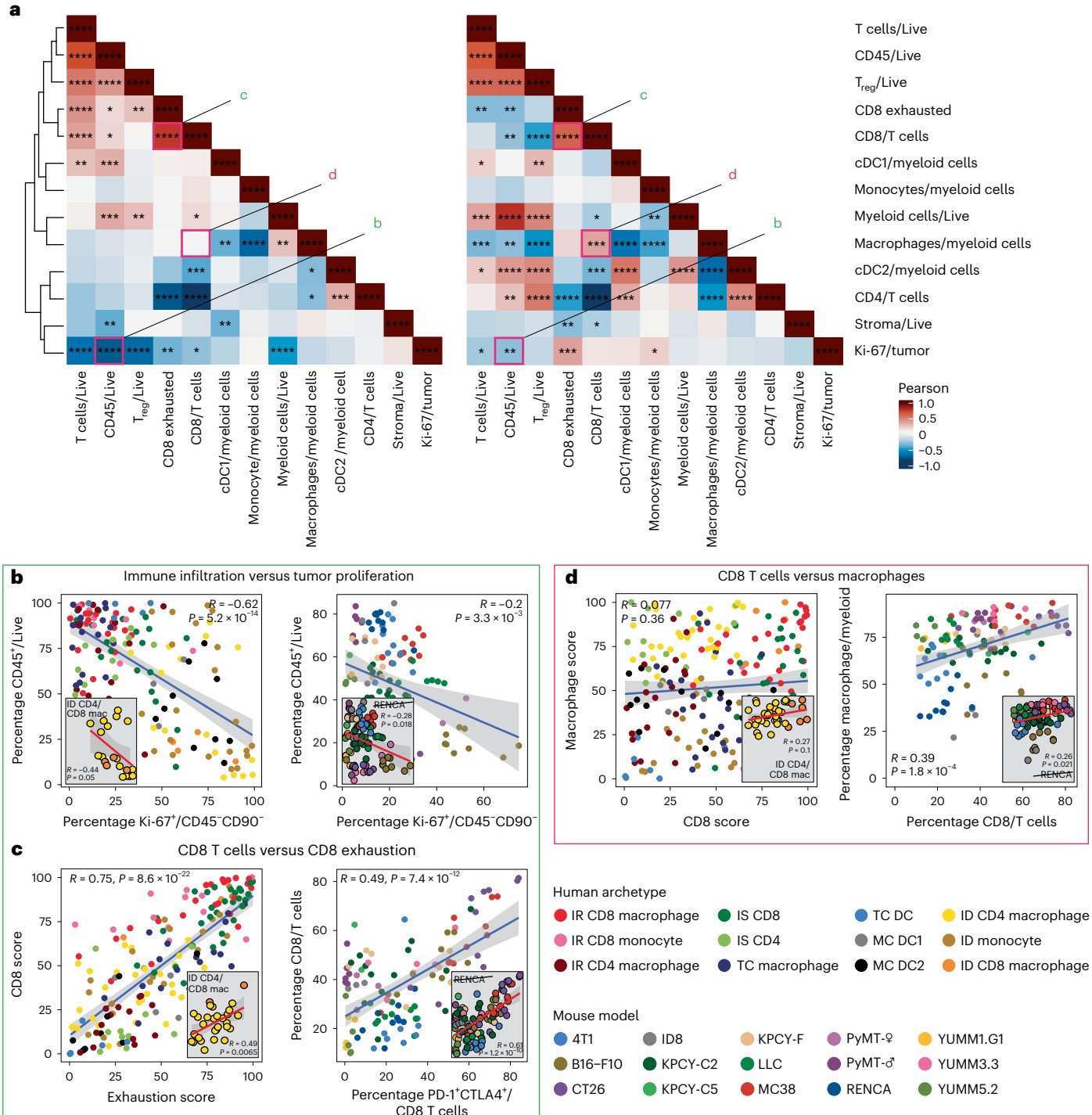

**Fig. 4 | Differential conservation of relative cell densities in the TME. a**, Pearson correlation matrices (human on the left, hierarchically clustered; mouse on the right, ordered according to the human matrix), presenting correlations between the frequencies of different immune components in the TME. **b**–**d**, Side-by-side dot plots exemplifying the Pearson correlations shown in **a** of immune parameters between human (left, colored according to archetypes) and murine (right, colored according to tumor line) tumors. The specific correlations are indicated above each panel and grouped as conserved correlation on the left (**b** and **c**) and nonconserved correlation on the right (**d**). mac, macrophage. The solid lines indicate the linear fit (lm) and the shaded ribbons show the standard errors of the fitted lines (s.e. of the fit). Statistical significance was calculated using a Pearson correlation test with Benjamini–Hochberg correction, *$P_{adj} \leq 0.05$, **$P_{adj} \leq 0.01$, ***$P_{adj} \leq 0.001$, ****$P_{adj} \leq 0.0001$. All statistical tests were two-sided; $P$ values are reported as exact values unless otherwise indicated.

Our data define metrics by which mouse models are well suited to study the immune response in macrophage-rich, poorly infiltrated tumor archetypes (Figs. 1 and 2) that span many tumor indications[7]. Several conserved relationships are intuitive, such as the correlation between T cell infiltration and T cell exhaustion in both species, suggesting that tumors can sustain high T cell densities only when those T cells are proportionally desensitized. Mouse models also reinforce our previous finding[7] that Ki-67 levels (a marker of proliferation, highest during the G2 phase of the cell cycle and degraded upon mitosis[64]) within tumor cells is inversely correlated with T cell infiltration

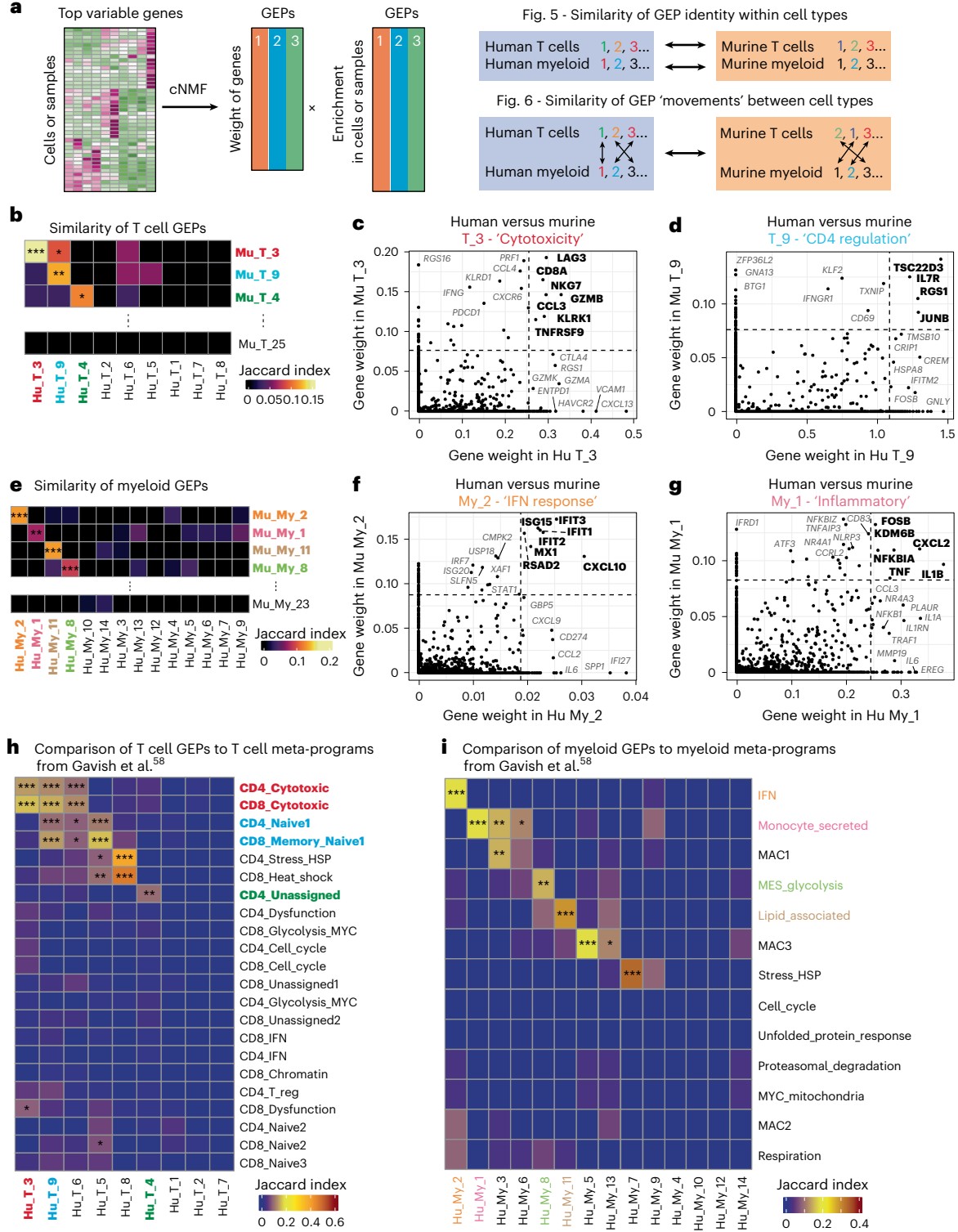

**Fig. 5 | Isolation of conserved and robust GEPs in T cells and myeloid cells.**
**a**, Schematic of our analytical strategy using cNMF to compare the identity and coordination of GEPs in T cells and non-granulocytic myeloid cells across species. **b**,**e**, Heatmaps showing the Jaccard indexes used to quantify the similarity between human and mouse GEPs in T cells (**b**, using the top 20 genes per GEP) and myeloid cells (**e**, using the top 50 genes per GEP). Similarities of interest are highlighted in bold, colored fonts. **c**,**d**,**f**,**g**, Scatter plots showing the gene contribution (that is, gene weight) to human versus murine GEPs T_3 (**c**), T_9 (**d**), My_2 (**f**) and My_1 (**g**). Genes in bold show the highest overlapping contributions

across species, while genes in gray have a higher contribution in one species versus the other. The dashed lines separate the 40 highest contributor genes from the others in either factor. **h**,**i**, Heatmaps showing the Jaccard indexes used to quantify the similarity between our human T cell (**h**) and myeloid cell (**i**) GEPs to the ones published in ref. 58. In **b**, **h** and **i**, statistical significance was calculated using a Fisher's exact test. *$P \le 0.05$, **$P \le 0.01$, ***$P \le 0.001$. All statistical tests were two-sided; $P$ values are reported as exact values unless otherwise indicated.

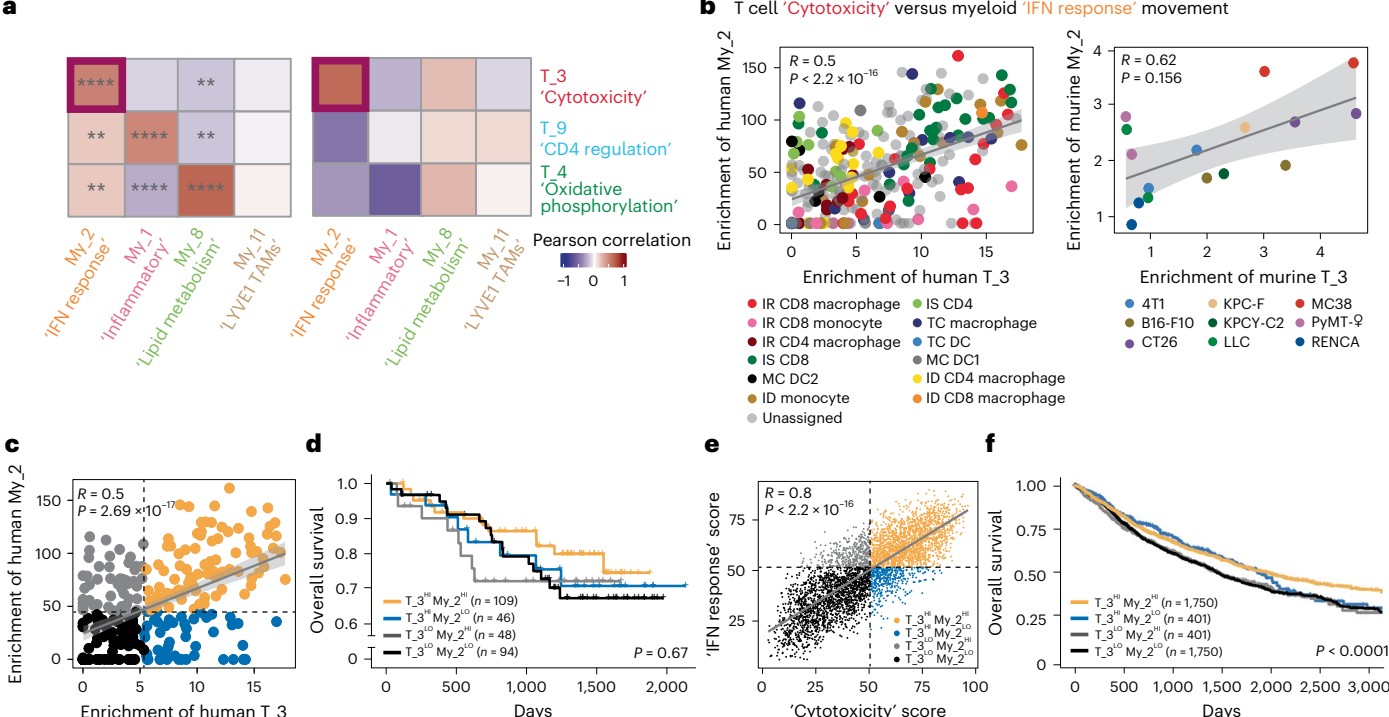

**Fig. 6 | Cross-species conserved GEP 'movements' between T cells and myeloid cells parses patients survival. a**, Heatmaps presenting the Pearson correlations of T cells versus myeloid GEP enrichment across human (left) or murine (right) TMEs. **b**, Scatter plots showing the correlation between the enrichments of GEPs T_3 and My_2 found across human (left, colored according to archetype) and mouse (right, colored according to tumor lines) tumors. Each dot represents a sample and the diagonal gray lines represent linear regressions. **c**, Scatterplot (corresponding to left of **b**) showing the binning of ImmunoProfiler patients as high or low for T_3 and My_2 GEPs, respectively (if present in the top or bottom 50% for each GEP enrichment). **d**, Kaplan–Meier graphs showing the overall survival of ImmunoProfiler patients stratified according to the binning shown in **c**. **e**, Scatterplot showing the relative enrichments of gene signatures calculated using the top 20 genes of GEPs T_3 and My_2 in patients from the TCGA. Patients were binned as high or low for each GEP if they were present in the top or bottom 50% for each calculated GEP gene score, respectively. Each dot represents a sample and the diagonal gray line represents the linear regression. **f**, Kaplan–Meier graph showing the overall survival of patients from the TCGA stratified according to the binning shown in **e**. In **b**, **c** and **e**, the solid lines indicate the linear fit (lm), the shaded ribbons show the s.e. of the fit. In **a**–**c** and **e**, statistical significance was calculated using a Pearson correlation test with Benjamini–Hochberg correction. *$P_{adj} ≤ 0.05$, **$P_{adj} ≤ 0.01$, ***$P_{adj} ≤ 0.001$, ****$P_{adj} ≤ 0.0001$. In **d** and **f**, statistical significance was calculated using a log-rank test. All statistical tests were two-sided; P values are reported as exact values unless otherwise indicated.

(Fig. 4). We speculate that this may highlight conserved tumor biology that fails to proteolyze Ki-67 after mitosis and is not allowing for high immune infiltration.

Furthermore, we find certain chemokine networks, including the *CXCR4* and *CXCR6* axes, to be strongly conserved from murine to human TMEs (Fig. 3). Other more complex themes exist across the spectrum of TMEs, including coordination of large GEP (Fig. 5), notably linking 'T cell cytotoxicity' and myeloid 'IFN response', that we and others have reported to be favorable to patient outcomes[65–67]. Specifically in the context of 'desert' human TME archetypes, murine models are especially relevant for studying population-level coordination, including correlations between macrophages and CD8 T cells (Fig. 4) and associated transcriptomic relationships between these cells.

In contrast, most mouse tumors fall short in modeling the T cell-dominant human TMEs that are common in kidney, lung or skin cancers[7]. This divergence extends beyond composition (Fig. 1 and also hinted in ref. 68) to differences in population coordination and transcriptomic regulation. This last point is exemplified by the differential coordination of T_9 'CD4 regulation' and My_1 'inflammatory' between human and murine TMEs (Fig. 6 and Extended Data Fig. 7), which warrants further examination as this myeloid GEP is associated with a poor outcome[62,63]. As we study this biology in greater detail, we note that patients who will benefit from cures derived in mice may be those with the worst current prognosis (Fig. 6) because they similarly have the mouse-like combination of low overall immune abundance (Fig. 1), high

macrophage fractions (Fig. 2), strong correlations between T cell and myeloid abundances (Fig. 4) and poorly coordinated 'CD4 regulation'/myeloid 'inflammatory' programs (Fig. 6 and Extended Data Fig. 7). This may offer opportunity in historically treatment-resistant cancers enriched for macrophage-dense, low T cell TMEs.

Among the 15 models analyzed, the RENCA model was a notable exception to our conclusions (as well as one of the ID8 samples, reflecting biological variation in this model when grown subcutaneously; Extended Data Fig. 1b). RENCA tumors are highly infiltrated by immune cells, among which we find the highest amounts of myeloid cells, dendritic cells and CD4 $T_{reg}$ cells among our murine TMEs cohort (Extended Data Fig. 1b). Therefore, RENCA seems to associate with human archetypes defined by a high myeloid content and biased toward CD4 T cells (IS CD4, MC DC1 and MC DC2[7]; Fig. 2b). We also find RENCA to be the lowest mouse models for T_3 'T cell cytotoxicity' and My_2 'IFN response' GEP enrichment (Fig. 6b), while being the highest enriched for T_9 'CD4 regulation' (Extended Data Fig. 7f). All these observations place RENCA as a model of choice to study immune populations and transcriptomic programs otherwise rarely found in other mouse tumors. Accordingly, our meta-analysis of 2,543 mouse samples further suggests that a minority of mouse tumors (~7%) can resemble highly infiltrated human subtypes (Fig. 2c). Thus, while rare, murine models capable of approximating immune-rich human TMEs exist.

The mechanistic basis for the relative paucity of T cells in murine tumors remains unclear. Possible explanations include evolutionary

differences in immune system prioritization (favoring innate versus adaptive responses) and reduced environmental immune stimulation in laboratory mice (by pathogens, commensals, immunizations). Although increasing microbial exposure to laboratory mice has been shown to better mimic human T cell responses[42,69,70], we did not find this type of environmental normalization to affect the T cell content of murine TMEs (Extended Data Fig. 1g). Also, while there is no evidence showing that human immune-rich tumors grow slower than immune-desertic tumors, the rapid growth kinetics of murine tumors may also limit adaptive responses in murine TMEs. We also note that our study focused on a single time point catching transitional stage tumors (day 14, most tumors measuring ~300–500 mm³) and therefore did not appreciate the dynamics of murine TME composition.

At present, our data do not explore the conserved or divergent mechanisms of response to immunotherapy between human and murine tumors. Given that responsive mouse models are often poorly infiltrated by T cells, yet macrophage-rich (for example, MC38, as well as CT26 and RENCA in some studies), and that human responses may involve both pre-existing and de novo T cell activity[71–73], benchmarking therapy-responsive states across species may reveal even larger discrepancies. Nonetheless, our framework using coordinated GEP 'movements' provides a strategy to evaluate conserved and divergent transcriptional circuits in response to perturbations[74], which is ongoing work.

Our study is not exhaustive and is enriched for syngeneic, transplanted models, with limited representation of orthotopic, spontaneous, genetically engineered or xenograft systems. Although preliminary observations in such models align with our broader conclusions (Extended Data Fig. 1f,g), additional analyses may identify murine tumors that better capture the diversity of human TMEs. This also applies to the cross-species comparisons between cell populations that our study did not assess, notably natural killer (NK) cells, B cells, neutrophils and innate lymphoid cells.

We also note that our cross-species GEP comparisons are influenced by differing platforms (bulk RNA-seq from cytometry-sorted populations in human versus scRNA-seq in mouse), potentially underestimating conservation (hinted at in Extended Data Fig. 7b) and limiting data interpretation. Furthermore, transcriptomic divergence such as differences in *CXCL13* expression or coordination between T_9 and My_1 GEPs may reflect species-specific regulatory architecture rather than absence of analogous biology[11,75,76].

## Conclusions

Typical mouse models incompletely represent the spectrum of human TMEs. While several cell–cell relationships (Fig. 4b,c) and core gene program linkages are conserved (Fig. 6b), important differences exist in immune composition (Figs. 1 and 2), chemokine networks (Fig. 3) and coordination of regulatory programs (Extended Data Fig. 7e,f). These findings underscore both the utility and the limits of murine models. Careful matching of mouse models to specific human TME archetypes (Fig. 4d and Extended Data Figs. 4e and 7e) and analysis of conserved transcriptional circuits will be essential for improving translational relevance while recognizing where divergence necessitates caution.

## Online content

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

## Methods

### Human ImmunoProfiler samples

Human samples were collected with patient consent after surgical resection under a UCSF institutional-review-board-approved protocol (UCSF number 20-31740), under the UCSF ImmunoProfiler project[7]. Briefly, freshly digested tumor samples were analyzed using flow cytometry or fluorescence-activated cell sorted into conventional T cell, $T_{reg}$, non-granulocytic myeloid, tumor and nonimmune CD44+CD90+ stroma compartments to perform bulk RNA-seq on individual cell fractions.

### Mouse tumor models

Unless specified, mice were housed at the AALAC-accredited animal facility of UCSF in specific pathogen-free (SPF) conditions. Animal experiments were approved and performed in accordance with the Institutional Animal Care and Use Committee protocol AN184232. For cell-line-based models, 6–8-week-old wild-type BALB/c (4T1, RENCA, CT26 models) or C57BL/6 (all other models) female mice were purchased from the Jackson Laboratory. Mice were euthanized at day 14 after tumor inoculation (typically ~300–500 mm³) for tumor collection and analyses. For the high-fat diet experiments, mice were fed either high-fat (Envigo TD.06414) or control (Envigo TD.08806) pellets for 2 weeks before starting the tumor experiments and until tumor collection. For the aging experiments, 90-week-old C57BL/6 female mice were purchased from the Jackson Laboratory and housed in our facility until reaching 2 years of age, when we started the experiments. The genetically engineered mouse model MMTV-PyMT-mChOVA develop spontaneous tumor lesions from 4–6 months of age, specifically in the mammary gland in females and in the salivary glands in males[77]. In these models, mice were euthanized and tumors collected when they reached ~300–500 mm³. In the experiments comparing mice housed under SPF or 'dirty' conditions[42], age-matched 6-week-old C57BL6/J female mice were purchased from The Jackson Lab and split into two cohorts, one housed under SPF conditions and the other co-housed with pet-store mice, therefore considered 'dirty'. Mice were housed/co-housed for 8-weeks before subcutaneous tumor cell injections of B16, MC38 or LLC, analysis of tumor growth, and endpoint analyses. Serological dosage of antibodies specific to various pathogens was used to confirm the 'dirty' status of the co-housed mice.

### Sample preparation

After collection, tumors were placed in a 12-well plate and minced to submillimeter pieces in 2 ml of Roswell Park Memorial Institute (RPMI) containing Collagenase IV (4 mg ml⁻¹) and DNase I (0.1 mg ml⁻¹). Tissues were then incubated at 37 °C for 30 min, with a mechanical dissociation step using thorough pipetting after the first 15 min. Digestion was stopped by adding 2 ml of cold RPMI + 10% FCS + 1× penicillin-streptomycin-glutamine to each sample before filtering through a 100-mm mesh and centrifugation at 500g for 5 min at 4 °C. Cells were then resuspended in RPMI + 10% FCS + 1× penicillin-streptomycin-glutamine for further counting and analysis.

### Mass cytometry

CyTOF was performed as described elsewhere[68]. Briefly, conjugations of mass cytometry antibodies with metal isotopes were done using the Maxpar conjugation kit (Fluidigm) according to the manufacturer's protocols and each antibody was titrated to define its optimal staining concentration. Each freshly digested sample was first stained with cisplatin, fixed in 3.2% paraformaldehyde (PFA) and frozen at −80 °C. Samples were then thawed and barcoded using mass-tag labeling with distinct combinations of stable Pd isotopes in PBS 0.02% saponin before pooling and staining. For this, cells were first resuspended in cell-staining medium (Fluidigm) containing metal-labeled antibodies against CD16 and CD32 for 5 min at room temperature to block Fc receptors, followed by the addition of a cocktail containing surface marker antibodies in a final volume of 500 µl for 30 min at room temperature. Cells were then permeabilized with methanol for 10 min at 4 °C, washed and incubated with a cocktail containing intracellular marker antibodies in a final volume of 500 µl for 30 min at room temperature (all antibodies are listed in Supplementary Table 3). Cells were finally stained with 191/193Ir DNA intercalator (Fluidigm) diluted in PBS with 1.6% PFA 48 h before data acquisition. For acquisition, cells were washed and resuspended at 1 M ml⁻¹ in deionized water + 10% EQ Four Element Calibration Beads (Fluidigm) and analyzed on a CyTOF mass cytometer (Fluidigm). We acquired an average of $1–3 \times 10^5$ cells per sample, which is consistent with accepted practices in the field. After data collection, we used the Premessa pipeline (https://github.com/ParkerICI/premessa) to normalize the data and deconvolute individual samples. We then manually gated the individual FCS files using FlowJo (Extended Data Fig. 1a).

### Immunofluorescence microscopy

For human samples, a 7-plex immunofluorescence panel was created using the Ventana BenchMark Ultra (Roche Diagnostics) automated staining platform from thin section of formalin-fixed paraffin-embedded blocks. All reagents were from Discovery (Ventana Medical Systems) and were used according to the manufacturer's instructions. Heat-induced epitope retrieval was performed with the Cell Conditioning 1 solution (catalog number 950-124) for 64 min at 97 °C. The primary antibodies used were CD3 (1:100 dilution, clone D7A6E, Cell Signaling Technology), CD4 (RTU, clone SP35, Ventana Medical Systems), CD8 (1:100 dilution, clone D8A8Y, Cell Signaling Technology), CD163 (1:250 dilution, clone EPR19518, Abcam), HLA-DR (1:500 dilution, clone EPR3692, Abcam), XCR1 (1:40 dilution, clone D2F8T, Cell Signaling Technology) and EpCAM (1:50 dilution, clone D9S3P, Cell Signaling Technology). The tissue was counterstained with 4′,6-diamidino-2-phenylindole (DAPI) (catalog number FP1490, Akoya Biosciences) for nucleus localization. Staining was conducted in two cycles: the first cycle had CD3, CD4, CD8, CD163, HLA-DR and XCR1; the second cycle had EpCAM. Both cycles used DAPI. The slide was scanned using a whole slide scanner after each staining cycle. Finally, the images from both cycles were registered to achieve the 7-plex image.

For the murine samples, B16-F10, MC38 and RENCA tumors were excised at day 14 after transplantation, fixed in 4% PFA for 4 days, dehydrated gradually in 15% and then in 30% sucrose before embedding in TissueTek OCT freezing medium (Sakura Finetek) and storage at −80 °C. Consecutive sections of 10-µm thickness were generated using a Leica CM3050 S cryostat. Sections were permeabilized and washed in PBS supplemented with 0.1% BSA and 0.1% Tween. Samples were blocked and stained in a blocking solution consisting of 0.5% TNB blocking reagent (Akoya Biosciences), 0.1 M Tris-HCl, 0.15 M NaCl, at pH 7.5. For staining, we used a cocktail containing anti-CD3e Alexa Fluor 647, anti-CD11b PE and anti-IA/IE Alexa Fluor 488 antibodies. Nucleus staining was performed using DAPI. Stained sections were finally mounted in Fluoromount G (Thermo Fischer Scientific) and analyzed on an inverted Leica Thunder microscope for tiled imaging using an HC PL FLUOTAR L ×20/0.4 numerical aperture objective. All images were acquired as tiled images, covering whole-tumor sections in the XY plane, with a single Z plane using the autofocus focal plane method in the Leica software.

### Image analysis

For both human and murine samples, images were processed and analyzed in QuPath[78]. Cells were segmented using StarDist[79] on DAPI. Classification of human cells as T (CD3+CD4+/−CD8+/−) or myeloid (CD3−HLA-DR+XCR1+/−CD163+/−) cells and classification of murine cells as

T cells (CD3⁺CD11b⁻MHC-II⁻) or myeloid cells (CD3⁻CD11b⁺ or MHC-II⁺) was done by training a random forest classifier in QuPath. For the human samples, we then calculated the T:Myeloid cell ratio for each patient (one slice stained per patient) and for mouse samples; we then calculated the T:Myeloid cell ratio for each slice, then averaged these ratios according to tumor sample (3–4 slices analyzed per tumor) and plotted the average ratio for each sample in a box plot using ggplot2 in R.

### Comparison of human and mouse TME diversity
Mouse CyTOF data and human flow cytometry data (inferred using linearity of the flow parameters with gene scores as shown in ref. 7) were used to generate immune feature matrices based on the ten PanCan immune features as described previously. After aggregating mouse samples according to tumor line and human samples according to immune archetype, the median frequency values for each immune feature were computed. Cosine similarity was computed by normalizing each row vector to unit length and taking the matrix product of the normalized data with its transpose. Cosine distance was then derived by subtracting the similarity values from 1.

### Immune subtype classification of mouse tumors
All RNA-seq samples in this analysis were derived from mouse tumors and collected from the NCBI Sequence Read Archive (SRA). To establish a cohort of mouse tumor samples, cancer-related disease ontologies (DOIDs) were gathered from the disease ontology database[80] using the 'ontologyIndex' package with the search terms 'cancer', 'carcinoma' and 'neoplasm', resulting in 1,529 unique DOIDs. MetaSRA[78,81], which maps SRA samples to terms in biomedical ontologies, was queried using the following parameters: 'RNA-seq', 'mouse', 'tissue' and the cancer-related disease ontology DOIDs. Samples with scRNA-seq ontologies were excluded. This query to MetaSRA returned 3,900 samples, with incomplete metadata and SRA sample IDs. Next, the MetaSRA results were filtered by manual examination of study metadata and abstracts to exclude samples not derived from tumors or using patient-derived xenografts, sorted cells or organoids, which resulted in a final cohort of 2,543 samples from 178 studies. RNA-seq gene counts for each study were accessed through recount3 (ref. 82), ensuring a uniform alignment pipeline and annotation records. Gene counts were normalized by dividing by the 75th percentile gene count for each sample and then applying a $\log_2$ transformation. Each mouse gene was converted to its human ortholog using the Babelgene package. To predict a tumor immune subtype for each sample, we used ImmuneSubtypeClassifier[83] (a machine-learning model comparing quantile and gene-pair features of 485 genes to determine immune subtypes[3]). Each sample was assigned its 'BestCall' subtype. A manual exploration of the metadata of the top contributor studies for this analysis showed that most (if not all) samples come from RNA-seq extracted from non-dissociated tumor tissues.

### Mouse scRNA-seq data generation
For most mouse experiment, we started by sampling $1 \times 10^6$ cells from the tumor of each animal and generated a single pool for each group. Therefore, a group of five mice generated a pool of $5 \times 10^6$ cells. This cell pool was then stained with the Zombie NIR viability dye (1/1,000 in PBS, 10 min at 4 °C) before being incubated with Fc block (clone 2.4G2, Tonbo Biosciences) and barcoded with HTO antibodies (TotalSeq-A, BioLegend). We then pooled all barcoded samples and stained them with fluorescently labeled antibodies (Supplementary Table 3). Using a BD FACSAria II Cell Sorter (BD Biosciences), we then gated live immune cells (Zombie-CD45⁺) and sorted two pools of cells from these samples: 'lymphoid cells' containing equal amounts of T cells (CD90.2⁺), B cells (B220⁺MHC-II⁺) and NK cells (CD49b⁺); and 'myeloid cells' gated as CD11b⁺ or CD11c⁺ among non-T, non-B and non-NK cells. These pools were then

individually encapsulated and complementary DNA libraries were built according to the 10X Genomics specifications for v.3 3′ chemistry. After fragment analysis and library quantification (BioAnalyzer), libraries were sequenced on an Illumina NovaSeq SP using 10X Genomics recommended sequencing parameters.

### Mouse scRNA-seq data processing
BCL files were converted to FASTQ format using cellranger mkfastq (v.3.0.2). FASTQ files were aligned to the GRCm38 genome, generating gene-by-cell count matrices with cellranger count. The resulting matrices were imported into Seurat (v.4.0.3) for downstream preprocessing and HTO demultiplexing, with manual inflection values provided to confidently retain singlet cells (https://github.com/UCSF-DSCOLAB/aarao_scripts/). After demultiplexing, quality control scores were calculated for mitochondrial, ribosomal and cell-cycling-related expression. Low-quality cells were defined as more than 10% mitochondrial RNA content, more than 60% ribosomal RNA content or fewer than 250 genes. Data were normalized (Seurat::NormalizeData), 3,000 variable features selected (Seurat::FindVariableFeatures) and then scaled (Seurat::ScaleData) with regression against percentage mitochondrial, percent ribosomal and cell cycle S and G2M scores. Principle components 1:30 were calculated (Seraut::RunPCA). Data were merged across samples and batch-corrected with (harmony::RunHarmony v.0.1.1), using batch as the grouping variable and retaining all default parameters for harmonization. UMAP (Seurat::RunUMAP) was then calculated using 30 principal components with harmony selected as the reduction. Finally, cell type annotation was performed by identifying cell clusters (Seurat::FindClusters) and grouping them based on their identify, inferred using differentially expressed genes (Seurat::FindAllMarkers with test.use = 'poisson' and latent.vars = 'Tumor_Line') and prior knowledge of cellular markers (Extended Data Fig. 2). During this process, we identified clusters of nonimmune cells, probably representing contamination during cell sorting, and containing both tumor and nonimmune stromal cells. These were present at stable ratios across all experiments; therefore, we decided to include the subpopulations representing PanCan-relevant tumor and stroma compartments[7] for the chemokines analyses of Fig. 3. The human-relevant mouse stroma compartment was identified by measuring the expression of the 20-gene human stroma signature from ref. 7 among all nonimmune cells and annotating high-scoring subsets as stroma in the mouse scRNA-seq data (Extended Data Fig. 2f).

### Chemokines analyses
To process the human bulk RNA sequencing data, raw counts were filtered as described previously[7]. Filtered counts were normalized to account for differences in sequencing depth via TMM normalization and converted to counts per million. Next, values were log-transformed and then $z$-scored across all compartments, enabling comparative analysis of chemokine expression across different compartments and patient archetypes. To better compare our scRNA-seq mouse data to the previously published bulk RNA sequencing data, the single-cell data were pseudobulked by calculating the average gene expression across each gene and condition; we then scaled the averaged expressions across all compartments. Finally, to identify homologous chemokines between mouse and human, the BioMart archive (April 2018) was used to convert mouse and human genes.

### Human scRNA-seq
The cohort of samples used in Fig. 3 to analyze population-specific expression of human chemokines ligands in human HNSC tumors has already been described elsewhere[84]. Briefly, ImmunoProfiler tumor samples were digested to single cells and then submitted to single-cell gene expression analyses through the cellular indexing of transcriptomes and epitopes by sequencing pipeline (Illumina) before analysis,

coarse annotation of TME compartments and visualization of gene expression across patients and TME compartments.

## cNMF

These analyses used a subset of the UCSF ImmunoProfiler Initiative dataset[7]. Bulk RNA-seq gene TPMs from sorted conventional T cells and non-granulocytic myeloid cells were filtered to include only samples from primary or metastatic tissue with a quality metric EHK score of 8 or greater (355 T cell samples and 322 myeloid samples). We applied $\log_2$ normalization to handle outliers and then filtered to the 5,000 most variable genes based on median absolute deviation. To obtain a quantitative measure of NMF rank stability, we used the cophenetic correlation coefficient as described in refs. [85,86]. We ran the NMF algorithm 50 times for each rank in the range of 3–30 factors. For each run, we generated a connectivity matrix, where the entries were 1 if the sample belonged to the same factor across runs and 0 if it did not. We then computed a consensus matrix by averaging all the connectivity matrices from the repeated runs. The cophenetic correlation coefficient (CCC), defined as the Pearson correlation between the consensus matrix and a sample distance matrix calculated using the Euclidean linkage method, was calculated for each rank. We repeated this process five times and plotted the median CCC against the rank. Ranks corresponding to local maxima in the median CCC plot were identified as stable candidate ranks for further analysis. To obtain the W and H matrices for the candidate ranks, we used a method similar to the single-cell workflow called DECIPHER-seq[57]. We ran the NMF algorithm ten times for each of the selected ranks using the Python module sklearn[41]. We then concatenated the W and H matrices from all ten runs to create larger matrices, $W_{10}$ and $H_{10}$. We then applied $k$-mean clustering with $k$ = candidate rank $H_{10}$ matrix and used the resulting cluster labels on the $W_{10}$ matrix. Next, we removed outliers in the clusters using the local outlier factor algorithm for anomaly detection, which computes the local density deviation of data points relative to their neighbors, applying a 40% cutoff. Removing outliers results in homogeneous clusters and helps with the interpretation of results downstream. After outlier removal, we calculated the median $W_{10}$ and $H_{10}$ to form the consensus matrices, $W_{Con}$ and $H_{Con}$.

For the murine dataset, cNMF was ran using DECIPHER-seq[57] on our mouse scRNA-seq data. Briefly, we isolated all conventional T and non-granulocytic myeloid cells and ran the DECIPHER-seq function 'iNMF_ksweep()' with the parameter batch = FALSE to run NMF on each population for 20 replicates of each $k$ from 2 to 40. Individual biological samples were used as the sample grouping variable.

## TCGA cohort and survival analyses

Analyses using the TCGA dataset were performed using the TCGA subcohort described in ref. 7. Briefly, tumor RNA-seq counts and TPMs along with curated clinical data for 13 cancer types (BLAD, CRC, GBM, gynecological (grouping ovarian, uterine corpus endometrial carcinoma and uterine carcinosarcoma), HNSC, KID, HEP, LUAD, PDAC, SARC and MEL) was filtered down to include primary solid tumors and metastatic samples only. This reduced the TCGA sample set to 4,341 tumor samples. GEP scores were generated by first normalizing (using percentiles) the expression values of the top 20 contributor genes for each GEP across all patients, followed by averaging these 20 normalized values for each patient. For survival analysis, patients were categorized as either high (top 30%) or low (bottom 30%) for each GEP score and analyzed using a log-rank test. The PanCan myeloid feature scores (relative to monocytes, macrophages, cDC1 and cDC2) were calculated in the same way, using the feature gene signatures from ref. 7.

## Reporting summary

Further information on research design is available in the Nature Portfolio Reporting Summary linked to this article.

## Data availability

Both human (https://quipi.org/app/quipi) and murine (https://quipi.org/app/quipi_humu) datasets can be readily queried and visualized using our user online interface. The human raw data can be accessed as described in ref. 7. The murine scRNA-seq data have been publicly deposited in the Gene Expression Omnibus under accession no. GSE310560. Source data are provided with this paper.

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

## Acknowledgements

We thank M. Spitzer (UCSF) and P. Turnbaugh (UCSF) for thoughtful discussions about the project. We thank E. Colisson (UCSF) and H. Jiang (UCSF) for providing the KPC tumors-bearing mice used in this study. We also thank I. Tenvooren (UCSF) and S. Tamaki (UCSF, Flow CoLab) for their assistance with CyTOF-related matters, R. Patel (UCSF, Data Science CoLab) for his input to data processing, and C. Chu and A. Poon (both from the UCSF Genomics CoLab) for their help with scRNA-seq quality control and libraries sequencing. Acquisition and analysis of certain human and murine samples was partially funded by AbbVie, Amgen, Bristol Myers Squibb, Genentech and Pfizer as part of the UCSF ImmunoProfiler Initiative (obtained by M.F.K.). Additional support came from the National Institutes of Health–National Cancer Institute (NIH–NCI; R01CA197363 and R21CA301376 to M.F.K.) and the UCSF ImmunoX Initiative (HuMu CoProject funding to M.F.K.).

## Author contributions

Conceptualization: T.C. and M.F.K. Experimentation: T.C., N.W.C., G.C.R., J.T., S.T., L.F.P., M.N. and N.V.G. Data processing and analysis: T.C., R.G.J., B.S., E.F., L.F.P., A.W.E., A.R., D.B., L.L.-J., J.P.G., D.A.S. and X.V. Data visualization: T.C., R.G.J., B.S. and H.W. Paper drafting: T.C., M.F.K., R.G.J. and B.S. Supervision: M.F.K., A.J.C. and G.K.F. External collaboration supervision: D.M. and E.T.L. Key internal collaborators: R.G.J., G.K.F. and A.J.C. Lead senior author: M.F.K.

## Competing interests

The authors declare no competing interests.

## Additional information

**Extended data** is available for this paper at

**Supplementary information** The online version
contains supplementary material available at

**Correspondence and requests for materials** should be addressed to
Tristan Courau or Matthew F. Krummel.

**Peer review information** *Nature Immunology* thanks the anonymous
reviewers for their contribution to the peer review of this work.
Peer reviewer reports are available. Primary Handling Editor: Stephanie
Houston, in collaboration with the *Nature Immunology* team.

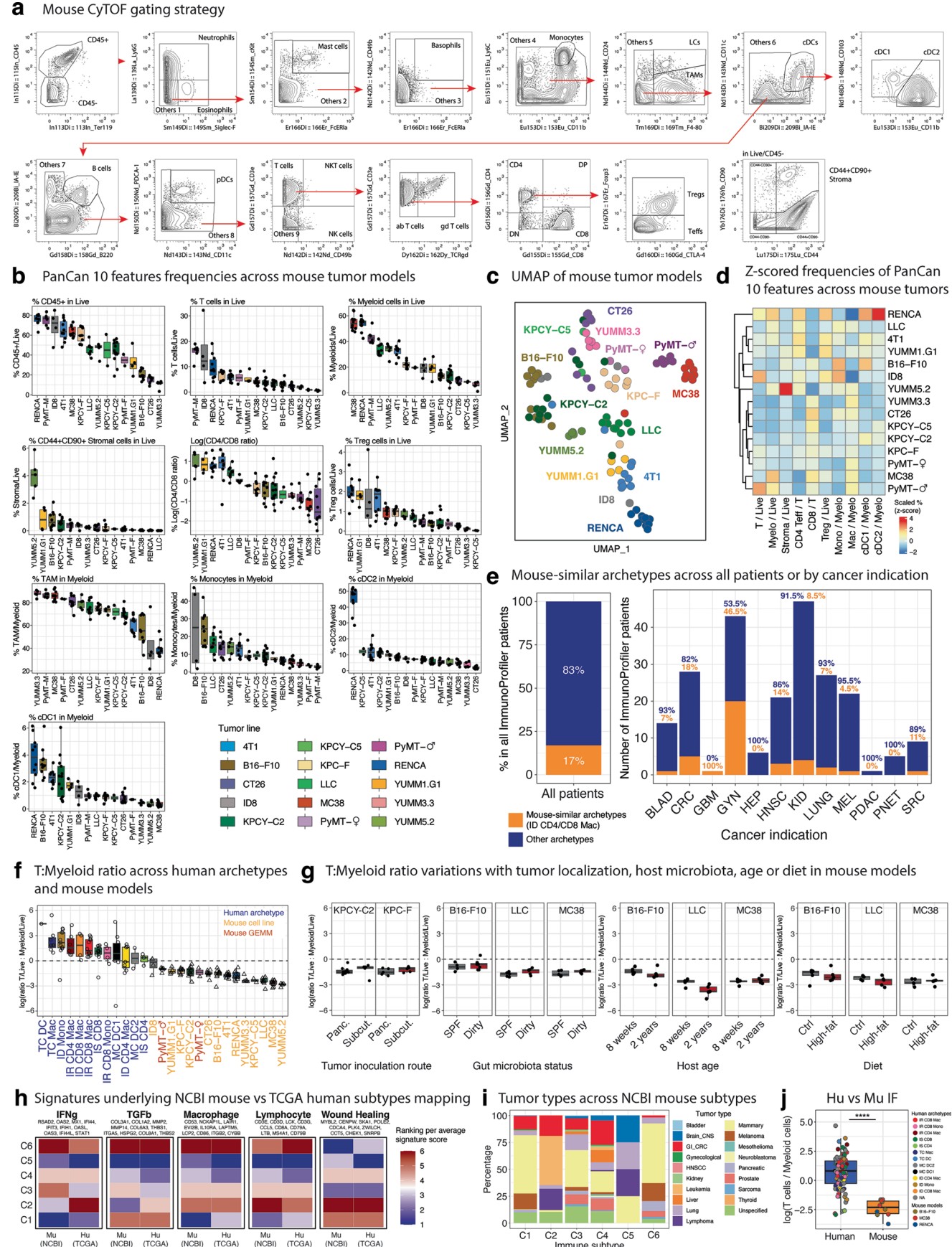

**a** Mouse CyTOF gating strategy

**b** PanCan 10 features frequencies across mouse tumor models

**c** UMAP of mouse tumor models

**d** Z-scored frequencies of PanCan 10 features across mouse tumors

**e** Mouse-similar archetypes across all patients or by cancer indication

**f** T:Myeloid ratio across human archetypes and mouse models

**g** T:Myeloid ratio variations with tumor localization, host microbiota, age or diet in mouse models

**h** Signatures underlying NCBI mouse vs TCGA human subtypes mapping

**i** Tumor types across NCBI mouse subtypes

**j** Hu vs Mu IF

**Extended Data Fig. 1 | See next page for caption.**

**Extended Data Fig. 1 | CyTOF gating strategy and diversity of immune profiles in muTMEs. a**. Gating strategy of immune populations using CyTOF analyses of mouse tumors. **b**. Waterfall boxplots describing the specific frequencies of the PanCan 10 features across mouse models. Data points represent individual mouse tumor samples. **c**. UMAP dimensionality reduction of individual mouse tumor samples, colored by model, using the 10 PanCan features. **d**. Heatmap presenting the hierarchical clustering of mouse tumors using the scaled PanCan 10 features. Scaling consisted in calculating the z-scores of each feature across mouse samples before averaging per model and then plotting values. **e**. Bar graphs presenting the frequencies of human samples whose TME composition can be mimicked by mouse models (in orange, i.e belonging to ID CD4/CD8 Mac archetypes) or not (in blue), either across the whole cohort (left) or split by tumor indication (right). **f** and **g**. Boxplots presenting the ratio of T cells over myeloid cells frequencies in Live across human and murine tumors (**f**.), between orthotopically (Panc.) vs subcutaneously (Subcut.) injected KPC pancreatic tumor models (left of **g**.) or in B16, LLC and MC38 tumors grown in mice kept under specific pathogen-free (SPF) vs dirty housing conditions (center-left of

**g**.), or grown in young (8-weeks) vs aged (2-years) mice (center-right of **g**.), or in mice fed with a control (Ctrl) vs High-fat diet (right of **g**.). **h**. Heatmap presenting scores of gene signatures derived from Thorsson et al. across human TCGA samples and the 2,543 murine NBCI samples used in Fig. 2c. For this, we first z-scored the expression of each gene for each signature across all TCGA patient. Then, for each signature we performed principal component analysis (PCA) and extracted the top genes contributing to PC1. PC1 weights were then applied to the z-score normalized mouse dataset and summed on a per-sample basis to calculate gene signature scores with the most human-relevant genes for each signature. Finally, we ranked the median scores from 1 to 6 in each species and plotted these ranks in a heatmap, order by immune subtypes calling as shown in Fig. 2c. The top contributing genes to each signature are labelled on the heatmap. **i**. Barplot showing the fractions of tumor types contributing to each immune subtype across the 2,543 murine NBCI samples. **j**. Similar boxplot than in Fig. 1f presenting the ratio of T cells over myeloid cells in human (n = 85) vs murine (n = 10) tumors, colored respectively by archetypes and tumor line.

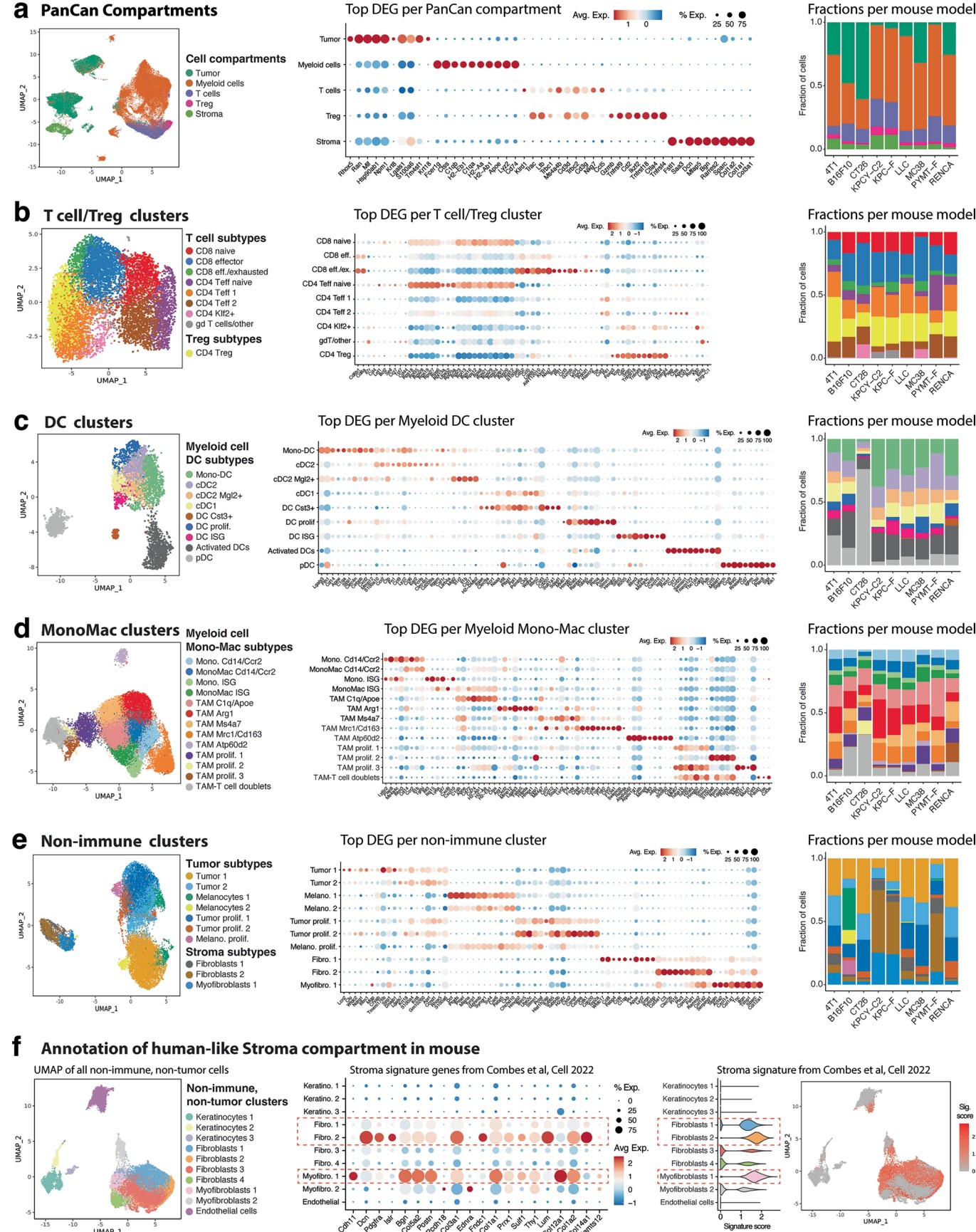

**Extended Data Fig. 2 | See next page for caption.**

**Extended Data Fig. 2 | Murine scRNAseq dataset. a**. UMAP visualization of 59551 cells merged from the entire mouse tumor scRNAseq data set (left), colored by their cellular compartment identity (i.e T cells, Treg, myeloid, tumor or stroma), as well as dotplot (middle) presenting the scaled expression of the top 10 differentially expressed genes (DEGs) between the 5 compartments and barplot (right) showing the fractions of each compartment per mouse model in the dataset. Note that the fractions in this barplot are influenced by our FACS sorting strategy prior to scRNAseq processing (as described in the methods) and therefore do not represent the real fraction of each compartment in the TME of each model. **b**. to **e**. As in a., UMAP plots (left), dotplots of top DEGs (middle) and barplots (right) describing the specific clusters of murine T cells (**b**.), DCs (**c**.), monocytes/macrophages (that is MonoMac, **d**.) and non-immune cells (**e**.). **f**. Description of the strategy followed to identify the human-like Stroma compartment in our mouse scRNAseq dataset using a UMAP (far left) presenting all non-immune cells identified in our murine dataset, a dotplot (middle left) showing the expression of the 20 genes composing the Stroma signature from[7] across our non-immune subsets, and a violin plot (middle right) and UMAP overlay (far right) showing the enrichment of a Stroma score (calculated using the 20 genes shown in the dotplot) in the non-immune cells. The specific cell subsets boxed in red in the dotplot and violin plot (Fibroblasts 1, 2 and Myofibroblasts 1) scored highly for the human Stroma genes and associated signature and were therefore considered as the human-relevant murine Stroma compartment for further analyses.

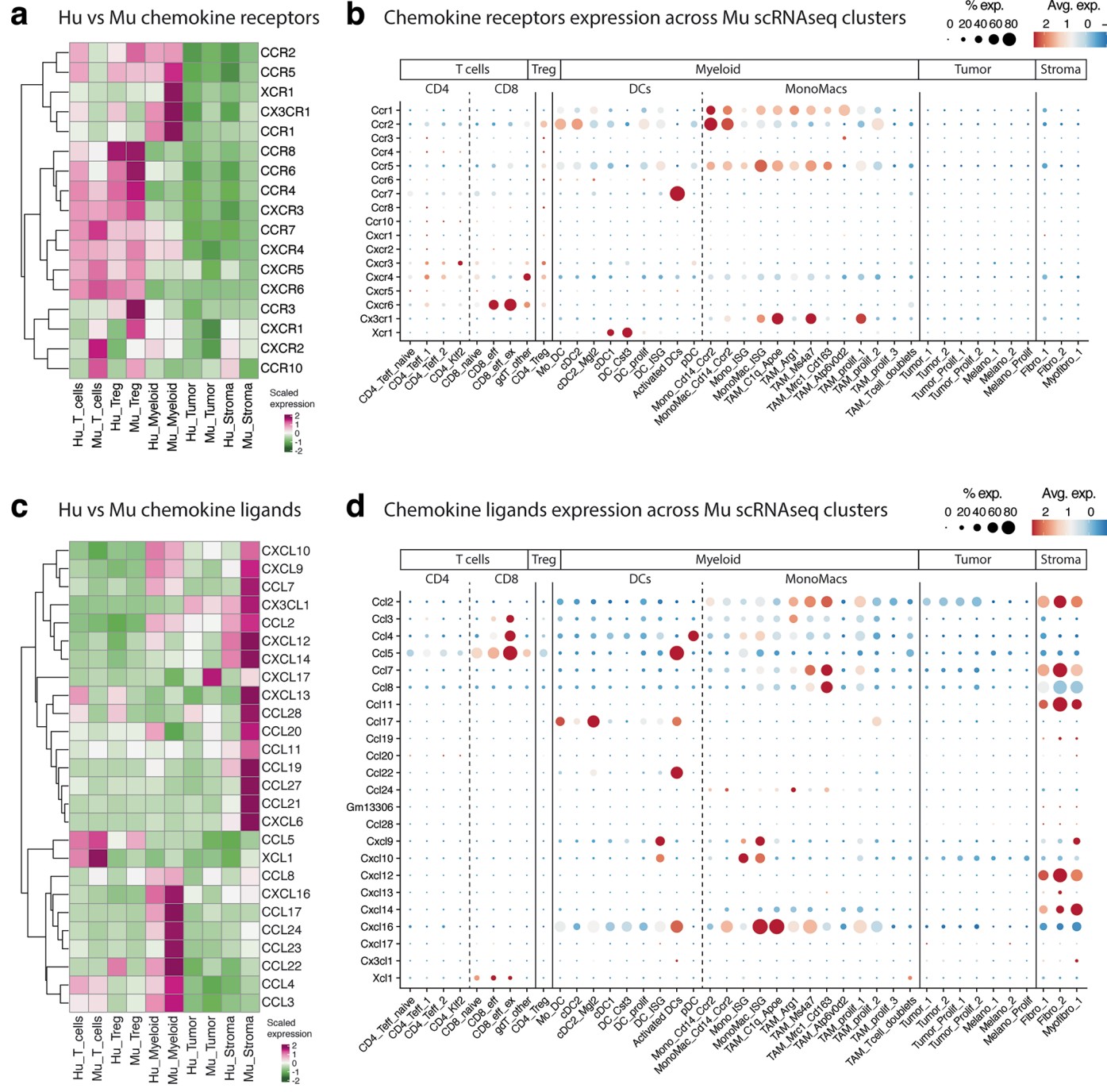

**Extended Data Fig. 3 | Additional chemokine pathways analyses. a. and c.** Heatmaps comparing the scaled expression of chemokines receptors (**a**.) and ligands (**c**.) between human and murine T cells, Treg, myeloid, tumor and stroma. For each species, the expression of each receptor/ligand was first scaled across compartments. We then extracted the scaled values for all human and murine compartments and plotted them side-by-side in a hierarchically clustered heatmap. **b. and d.** Dotplots presenting the scaled expression of the chemokine receptors (**b**.) and ligands (**d**.) shown in **a.** and **c.** across the murine cell clusters from scRNAseq shown in Extended Data Fig. 2b to e.

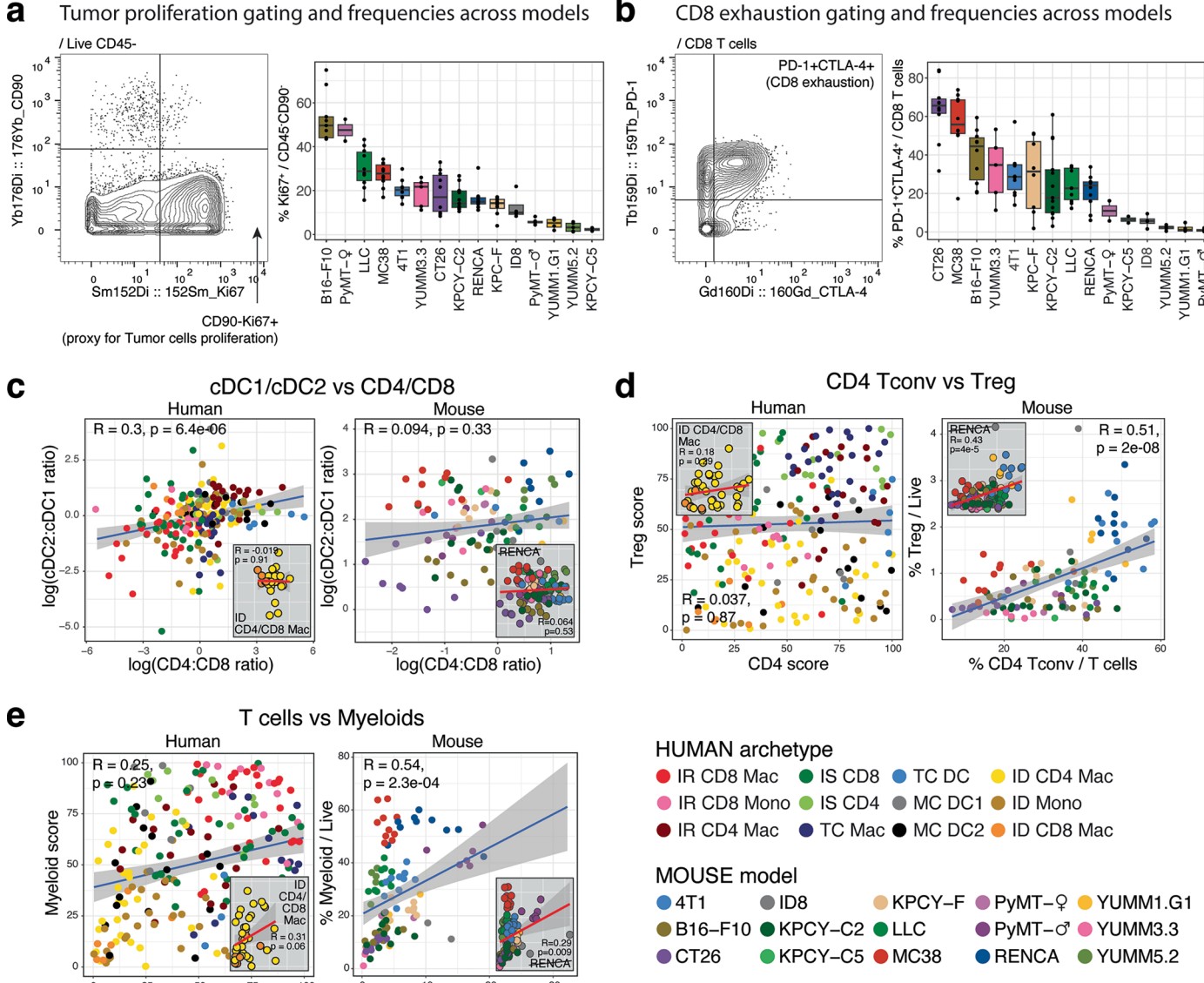

**Extended Data Fig. 4 | Additional examples of immune frequencies coordination. a. and b.** Gating strategy (left) and boxplot (right) presenting the frequencies of tumor proliferation (defined as Ki67+ in CD45−CD90− cells, **a.**) and CD8 exhaustion (defined as PD-1+CTLA-4+ in CD8 T cells, **b.**) in mouse

tumor samples. **c.** to **e.** Dot plots exemplifying some correlations shown in Fig. 4a. between human (left, colored by archetypes) vs murine (right, colored by tumor line) tumors. Statistical significance in **c.** to **e.** was calculated using a Pearson correlation test with Benjamini-Hochberg correction.

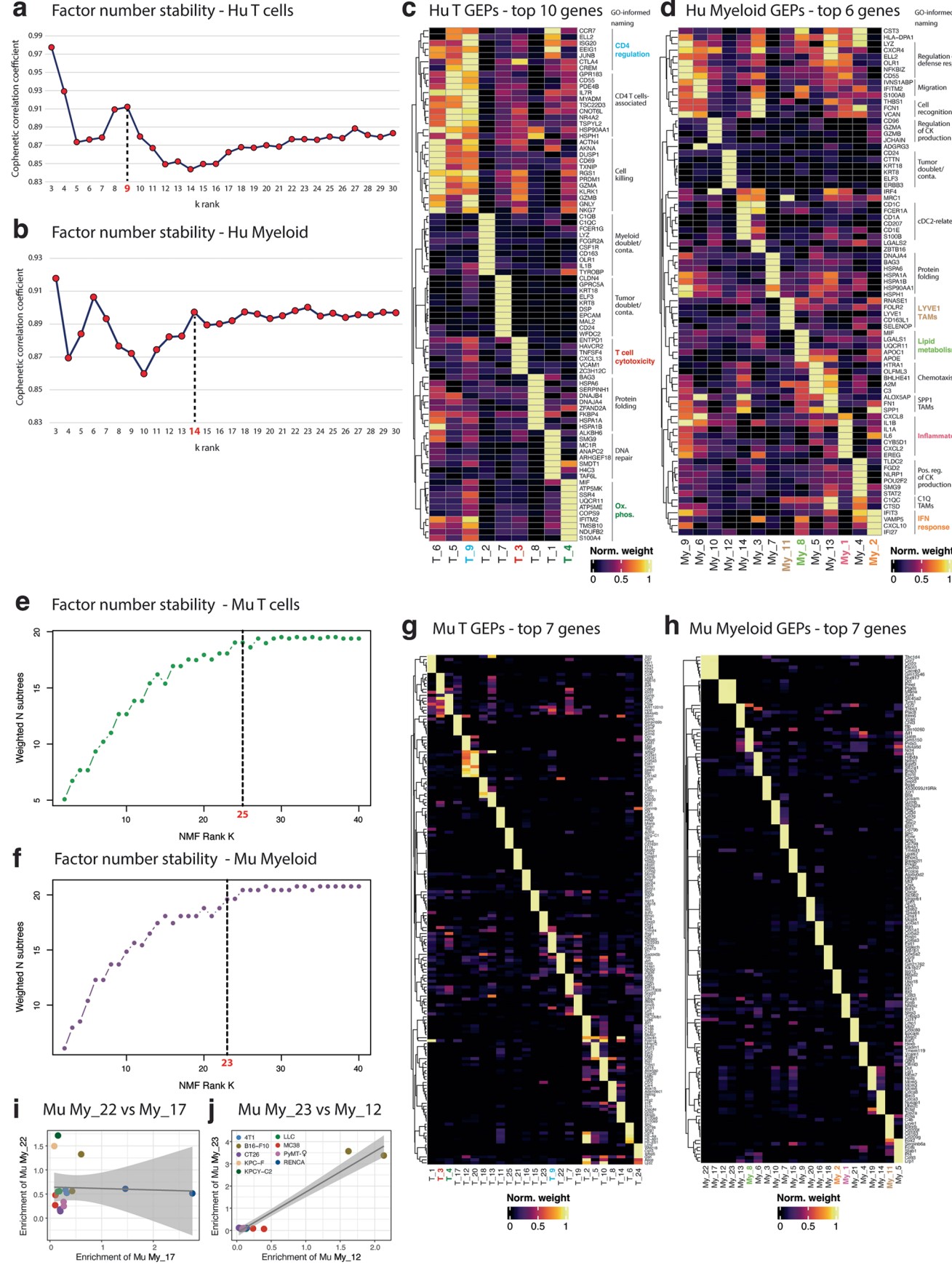

**Extended Data Fig. 5 | See next page for caption.**

**Extended Data Fig. 5 | Identification of T cells/myeloid gene expression programs (GEPs) using cNMF. a. and b.** Scatter plots showing the cophenetic correlation coefficients (CCC) used to establish the stability of each cNMF resolution in human T cells (**a.**) and myeloid cells (**b.**). We picked the number of stable GEPs (in red) as the value associated to a peak in CCC followed by a flattening of the curve. **c. and d.** Heatmaps showing the scaled weights (scaling first all genes in each GEP and then each scaled gene across all GEPs) of the top genes composing T cells (**c.**) or myeloid (**d.**) GEPs. **e. and f.** Scatter plots showing the weighted N subtrees (WNS) values used to establish the stability of each cNMF resolution in mouse T cells (**e.**) and myeloid (**f.**). We picked the number of stable GEPs (in red) as the value associated to the highest WNS value that was followed by a flattening of the curve. **g. and h.** Heatmaps as in c. and d. showing the scaled weights of the top genes composing mouse T cells (**h.**) and myeloid (**h.**) GEPs. **i. and j.** Scatter plots showing the relative enrichments of GEPs My_17 vs My_22 (**i.**) and My_12 vs My_23 (**j.**) across mouse samples, colored by tumor model. Note that among the 23 murine myeloid GEPs we noticed 2 apparent factors duplication, namely My_22/My_17 and My_23/My_12. While My_22 and My_17 seem to share gene composition (related to activated DCs), they are enriched in different sets of samples (**i.**). However, My_23 and My_12 appear as a duplication that could arise from over clustering of the NMF algorithm, because their gene composition shows a low degree of contamination/doublet with melanocytes and they are both uniquely found in B16-F10 samples (**j.**).

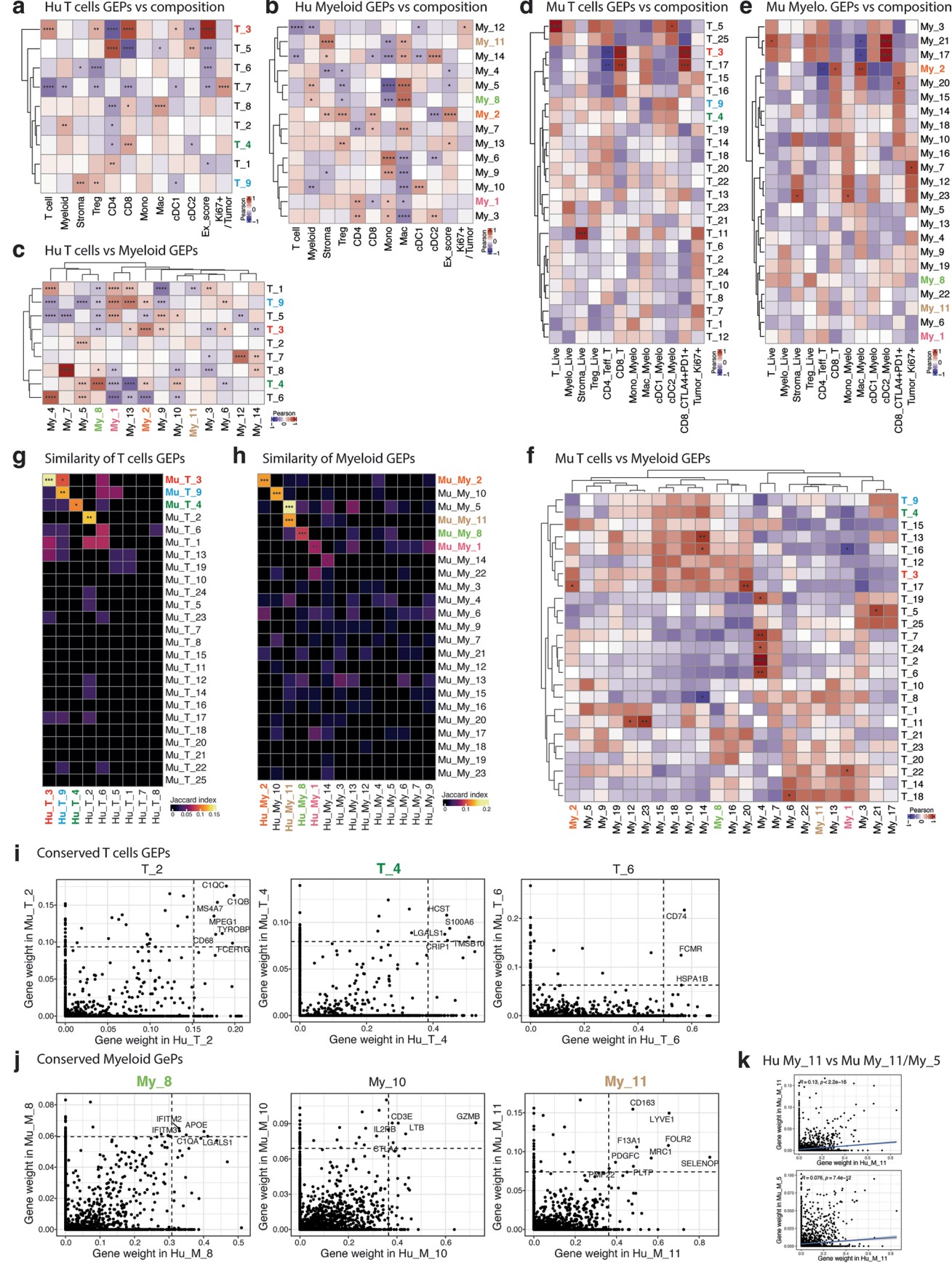

**Extended Data Fig. 6 | See next page for caption.**

**Extended Data Fig. 6 | Conservation of human vs murine GEPs, correlations with tumor immune composition and coordinated T vs myeloid GEPs 'movements'. a. and b.** Heatmaps presenting the Pearson correlations of either T cells (**a.**) or myeloid (**b.**) GEPs' enrichment vs the immune composition of human tumor samples. **c.** Heatmaps presenting the Pearson correlations of T cells vs myeloid GEPs' enrichment across human tumor samples. **d., e.** and **f.** Heatmaps displaying the same Pearson correlations as in **a.**, **b.** and **c.**, respectively, across mouse tumor samples. **g.** and **h.** Heatmaps (non-truncated) showing the Jaccard indexes used to quantify the similarity between human vs mouse GEPs in T cells (**g.**, using top 20 genes per GEP) and myeloid cells (**h.**, using top 50 genes per GEP). Note that although GEPs T_2, T_6 and My_10 appeared similar between mouse and human, we removed them from further analyses as they likely represent a low degree of myeloid contamination/doublets present in T cells for T_2 (Supplementary Table 2), or conversely for My_10 (Supplementary Table 2), or show poor overlap of top driver genes for T_6 (Extended Data Fig. 6i and Supplementary Table 2). **i.** and **j.** Scatter plots showing the genes contribution

(i.e genes weight) to human vs murine conserved T cells (**i.**) or myeloid (**j.**) GEPs. Genes annotated show the highest overlapping contributions across species, and the dashed lines separate the 40 highest contributor genes from the others in either factor. **k.** Scatter plots showing the Pearson correlations of the genes weights between human My_11 vs either murine My_11 (left) or murine My_5 (right). Linear regression, R and p values are displayed. Note that both murine My_5 and My_11 show similarity with human My_11 (**h.**). As these 2 factors seem to be driven by different genes (Extended Data Fig. 5h), we evaluated the correlation of their specific genes' weights with the ones of Hu_My_11 (**k.**) to show that Mu_My_11 seem to more closely mimic both the composition (among which LYVE1, FOLR2, CD163, MRC1 and SELENOP) and relative gene weights of Hu_My_11. Statistical significance in **a.** to **c.** and **b.** was calculated using a Pearson correlation test with Benjamini-Hochberg correction, * p.adj ≤ 0.05, ** p.adj ≤ 0.01, *** p.adj ≤ 0.001, **** p.adj ≤ 0.0001. Statistical significance in **g.** and **h.** was calculated using a Fisher's exact test. * p ≤ 0.05, ** p ≤ 0.01, *** p ≤ 0.001.

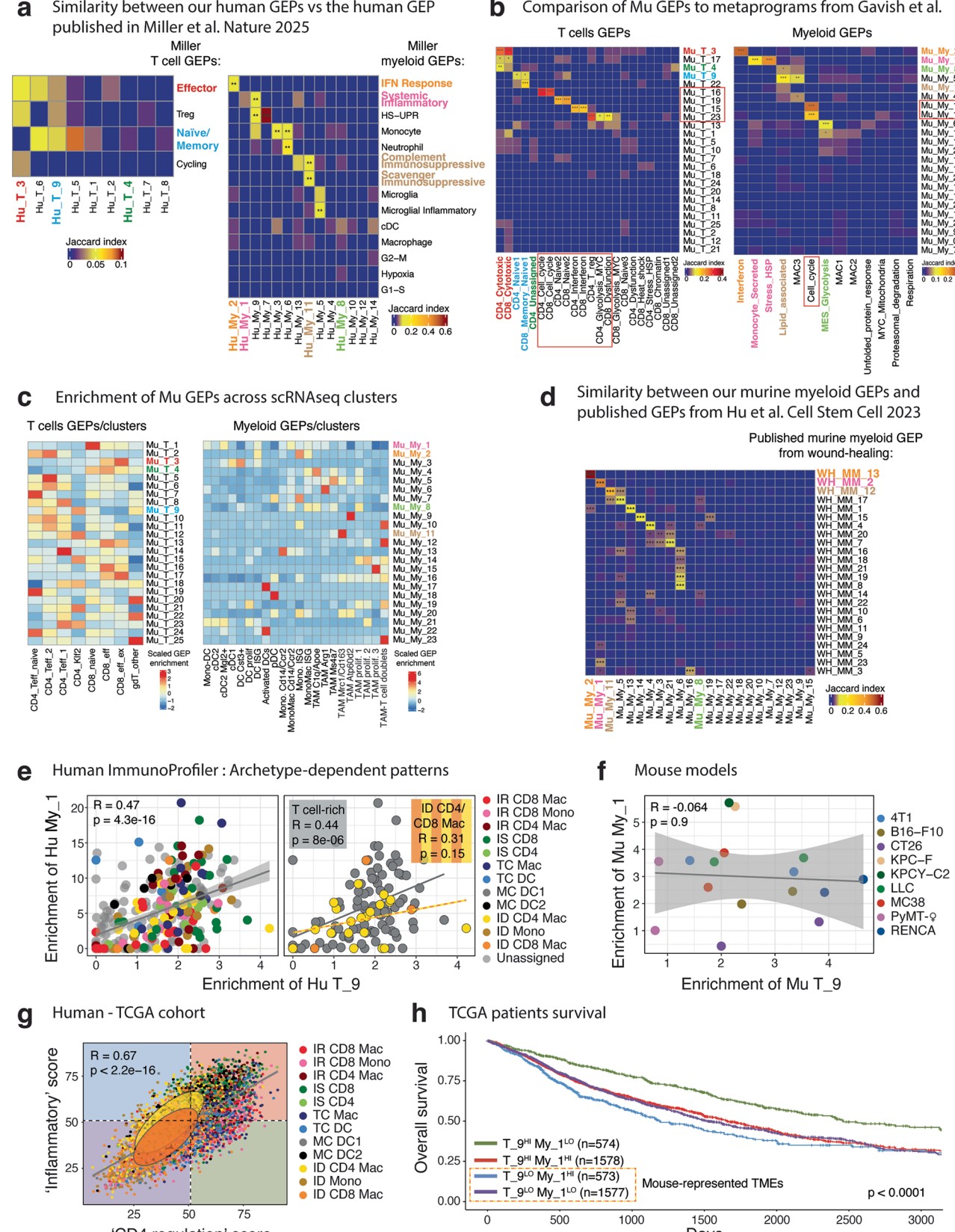

**Extended Data Fig. 7 | See next page for caption.**

**Extended Data Fig. 7 | Robustness of our T cells and myeloid GEPs and learning from the discordance of their 'movements' between huTMEs vs muTMEs.**
**a. and d.** Heatmaps showing Jaccard indexes used to quantify the similarity between (**a.**) our human T cells (left, using top 20 genes per GEP) and myeloid (right, using top 50 genes per GEP) GEPs to the ones published in[59] or between (**d.**) our murine myeloid GEPs and the ones published in[24] (using top 50 genes per GEP). The top 20 genes for each GEP were used to calculate similarity.
**b.** Heatmaps showing the Jaccard indexes used to quantify the similarity between our scRNAseq-based mouse GEPs and the scRNAseq-based GEPs published in[59] for T cells (left) and myeloid cells (right). Similarities of interest are highlighted in bold, colored fonts. **c.** Heatmaps presenting the enrichment of mouse GEPs in each cell subtype shown in Extended Data Fig. 2 (T cells on the left and myeloid cells on the right). GEPs enrichment values (H values) were scaled across all individual cells and then averaged by cell subtype before plotting. **e.** and **f.** Scatter plots showing the correlation between the enrichments of GEPs T_9

and My_1 found across human (**e.**, colored by archetypes) and mouse tumors (**f.**, colored by tumors lines). Each dot represents a sample, and the diagonal grey lines represent linear regressions. **g.** Scatter plot showing the relative enrichments of gene signatures calculated using the top 20 genes of GEPs T_9 and My_1 in TCGA patients. Patients were binned as High or Low for each GEP if they were present respectively in the top or bottom 50% for each calculated GEP gene score. Each dot represents a sample, and the diagonal grey line represents linear regression. The yellow and orange ellipses highlight the behavior of patients belonging to the Immune Desert CD4/CD8 Macrophages archetypes. **h.** Kaplan-Meier graph showing the overall survival of TCGA patients stratified according to their respective enrichment of T_9 and My_1 following the binning shown in **e.** Statistical significance in **a.** to **d.** was calculated using a Fisher's exact test. Statistical significance in **e.** to **g.** was calculated using a Pearson correlation test with Benjamini-Hochberg correction. Statistical significance in h. was calculated using a log-rank test.

# Reporting Summary

## Statistics

For all statistical analyses, confirm that the following items are present in the figure legend, table legend, main text, or Methods section.

| n/a | Confirmed | |
|---|---|---|
| ☐ | ☒ | The exact sample size (*n*) for each experimental group/condition, given as a discrete number and unit of measurement |
| ☐ | ☒ | A statement on whether measurements were taken from distinct samples or whether the same sample was measured repeatedly |
| ☐ | ☒ | The statistical test(s) used AND whether they are one- or two-sided *Only common tests should be described solely by name; describe more complex techniques in the Methods section.* |
| ☒ | ☐ | A description of all covariates tested |
| ☐ | ☒ | A description of any assumptions or corrections, such as tests of normality and adjustment for multiple comparisons |
| ☐ | ☒ | A full description of the statistical parameters including central tendency (e.g. means) or other basic estimates (e.g. regression coefficient) AND variation (e.g. standard deviation) or associated estimates of uncertainty (e.g. confidence intervals) |
| ☐ | ☒ | For null hypothesis testing, the test statistic (e.g. *F*, *t*, *r*) with confidence intervals, effect sizes, degrees of freedom and *P* value noted *Give P values as exact values whenever suitable.* |
| ☒ | ☐ | For Bayesian analysis, information on the choice of priors and Markov chain Monte Carlo settings |
| ☒ | ☐ | For hierarchical and complex designs, identification of the appropriate level for tests and full reporting of outcomes |
| ☐ | ☒ | Estimates of effect sizes (e.g. Cohen's *d*, Pearson's *r*), indicating how they were calculated |

*Our web collection on statistics for biologists contains articles on many of the points above.*

## Software and code

Policy information about availability of computer code

| Data collection | BD LSR Fortessa X20 (BD Biosciences), LSRFortessa (BD Biosciences), CyTOF 2 mass cytometer (Fluidigm), NovaSeq (Illumina) were used for data collection |
|---|---|
| Data analysis | Data analysis tools are described in the methods section of our manuscript. Flow cytometry data was collected using BD FACSDiva software and analyzed with FlowJo (BD). CyTOF data was collected using a CyTOF 2 Fluidigm mass cytometer, debarcoded and normalized in R using Premessa before analysis using FlowJo (BD). BulkRNAseq of human samples was described in Combes et al. Cell 2022. Single-cell analysis of human samples was described in Ray et al. Sci Immunol 2025. Single-cell analysis of mouse samples was performed using cellranger, Seurat and ggplot2 in R. |

For manuscripts utilizing custom algorithms or software that are central to the research but not yet described in published literature, software must be made available to editors and reviewers. We strongly encourage code deposition in a community repository (e.g. GitHub). See the Nature Portfolio guidelines for submitting code & software for further information.

## Data

Policy information about availability of data

All manuscripts must include a data availability statement. This statement should provide the following information, where applicable:
- Accession codes, unique identifiers, or web links for publicly available datasets
- A description of any restrictions on data availability
- For clinical datasets or third party data, please ensure that the statement adheres to our policy

Both human (https://quipi.org/app/quipi) and murine (https://quipi.org/app/quipi_humu) datasets can be readily queried and visualized using our user online interface. The Human raw data can be accessed as described in Combes et al. Cell 2022. The murine scRNAseq data has been publicly deposited in GEO under accession number GSE310560.

## Research involving human participants, their data, or biological material

Policy information about studies with human participants or human data. See also policy information about sex, gender (identity/presentation), and sexual orientation and race, ethnicity and racism.

| | |
|---|---|
| Reporting on sex and gender | Patient metadata recorded (described in Combes et al, Cell 2022) but not used to make comparisons based on sex and gender |
| Reporting on race, ethnicity, or other socially relevant groupings | Patient metadata recorded (described in Combes et al, Cell 2022) but not used to make comparisons based on race, ethnicity or other socially relevant groupings |
| Population characteristics | Patient metadata recorded and described in Combes et al, Cell 2022 |
| Recruitment | Described in Combes et al, Cell 2022 |
| Ethics oversight | All human tumor samples were collected with patient consent after surgical resection under a UCSF IRB approved protocol (UCSF IRB# 20-31740), under the UCSF ImmunoProfiler project as described in Combes et al, Cell 2022. |

Note that full information on the approval of the study protocol must also be provided in the manuscript.

# Field-specific reporting

Please select the one below that is the best fit for your research. If you are not sure, read the appropriate sections before making your selection.

☒ Life sciences      ☐ Behavioural & social sciences      ☐ Ecological, evolutionary & environmental sciences

For a reference copy of the document with all sections, see nature.com/documents/nr-reporting-summary-flat.pdf

# Life sciences study design

All studies must disclose on these points even when the disclosure is negative.

| | |
|---|---|
| Sample size | We used 2-14 mice per experimental group for all studies. A precise list of animal numbers per group per analysis is provided as a supplementary table with the manuscript. No statistical methods were used to pre-determine sample sizes. |
| Data exclusions | No data were excluded. |
| Replication | All mass cytometry and scRNAseq experiments were performed twice with multiple biological independent samples. Data was pooled as indicated in the relevant supplementary table. |
| Randomization | No randomization was needed in this study |
| Blinding | No blinding was done for this study |

# Reporting for specific materials, systems and methods

We require information from authors about some types of materials, experimental systems and methods used in many studies. Here, indicate whether each material, system or method listed is relevant to your study. If you are not sure if a list item applies to your research, read the appropriate section before selecting a response.

## Materials & experimental systems

| n/a | Involved in the study |
|-----|----------------------|
| ☐ | ☒ Antibodies |
| ☐ | ☒ Eukaryotic cell lines |
| ☒ | ☐ Palaeontology and archaeology |
| ☐ | ☒ Animals and other organisms |
| ☒ | ☐ Clinical data |
| ☒ | ☐ Dual use research of concern |
| ☒ | ☐ Plants |

## Methods

| n/a | Involved in the study |
|-----|----------------------|
| ☒ | ☐ ChIP-seq |
| ☐ | ☒ Flow cytometry |
| ☒ | ☐ MRI-based neuroimaging |

## Antibodies

| Antibodies used | Mouse CyTOF antibodies:<br>Specificity Supplier Reference Clone<br>B220 Biolegend 103202 RA3-6B2<br>CCR7 Biolegend 120101 4B12<br>CD103 Biolegend 121402 2E7<br>CD11b Biolegend 101202 M1/70<br>CD11c Biolegend 117302 N418<br>CD16/32 BD 553142 2.4G2<br>CD206 Biolegend 141702 C068C2<br>CD24 Biolegend 101802 M1/69<br>CD3e Biolegend 100202 17A2<br>CD301b Biolegend 146802 URA-1<br>CD38 Biolegend 102702 90<br>CD4 Biolegend 100506 RM4-5<br>CD44 Biolegend 103002 IM7<br>CD45 Biolegend 103102 30-F11<br>CD49b Biolegend 103513 HMa2<br>CD62L R&D MAB5761 MAB5761<br>CD64 Biolegend 139302 X54-5/7.1<br>CD69 R&D AF2386 Polyclonal<br>CD8 Biolegend 100702 53-6.7<br>CD86 Biolegend 105002 GL-1<br>CD90 Biolegend 105202 G7<br>cKit Biolegend 105802 2B8<br>CTLA-4 Biolegend 106302 UC10-4B9<br>F4/80 Biolegend 123102 BM8<br>FcER1a Biolegend 134302 MAR-1<br>Flt3 eBiosciences 14-1351-85 A2F10<br>Foxp3 eBiosciences 14-4771-80 NRRF-30<br>GATA3 Biolegend 653802 16E10A23<br>ICOS Biolegend 313502 C398.4A<br>Ki67 eBiosciences 14-5698-82 SolA15<br>Ly6C Biolegend 128002 HK1.4<br>Ly6G Biolegend 127602 1A8<br>MHC II Biolegend 107602 M5/114.15.2<br>PD-1 Biolegend 135202 29F.1A12<br>PD-L1 Biolegend 124302 10F.9G2<br>PDCA-1 Biolegend 127002 927<br>RORgt eBiosciences 14-6981-82 B2D<br>Siglec-F BD 552125 E50-2440<br>Siglec-H Biolegend 129602 551<br>SIRPa Biolegend 144002 P84<br>T-bet Biolegend 644802 4B10<br>TCRgd Biolegend 118101 GL3<br>Ter119 Biolegend 116202 Ter119<br>Tim-3 Biolegend 134002 B8.2C12<br><br>Mouse scRNAseq sorting panel:<br>Specificity Fluorochrome Supplier Clone Reference<br>B220 Alexa Fluor 488 Biolegend RA3-6B2 103225<br>CD49b PE Biolegend HMα2 103506<br>I-A/I-E Brilliant Violet 421 Biolegend M5/114.15.2 107632<br>CD11b Brilliant Violet 510 Biolegend M1/70 101245<br>CD11c Brilliant Violet 650 Biolegend N418 117339<br>CD90.2 Brilliant Violet 785 Biolegend 30-H12 105331<br>CD45.2 Brilliant Ultra Violet 395 BD 104 564616 |
|-----------------|---|
| Validation | All antibodies conjugated to fluorophores are commercially available and validated both by the manufacturer and through citations |

| Validation | in the scientific literature. Validation materials for each antibody are accessible on the respective manufacturer's homepage. |
|---|---|
| | For mass cytometry, antibodies were purchased unlabeled and conjugated to heavy metals in-house. Antibody conjugation to heavy metal tags was done using the MaxPar Antibody Conjugation Kit (Fluidigm) according to the manufacturer's protocol. After labeling, antibodies were diluted to 0.2mg-0.5mg/mL in antibody stabilization solution (Candor Bioscience) and stored at 4°C until use. Before using experimentally, conjugated antibodies were titrated on mouse tissue to determine optimal staining concentration. |

# Eukaryotic cell lines

Policy information about cell lines and Sex and Gender in Research

| Cell line source(s) | Cell line Origin Culture media Incubation Injection site # injected<br>B16-F10 ATCC CRL-6475 DMEM (Gibco 11995-065), 10% FCS (Benchmark), 1X Penicillin-Streptomycin-Glutamine (ThermoFisher) 37ºC 5% CO2 Subcutaneous, right flank 250K<br>LLC ATCC CRL-1642  37ºC 5% CO2 Subcutaneous, right flank 500K<br>MC38 Sigma-Aldrich SCC172  37ºC 5% CO2 Subcutaneous, right flank 500K<br>4T1 ATCC CRL-2539  37ºC 5% CO2 Mammary fat pad 250K<br>YUMM1.G1 ATCC CRL-3363  37ºC 5% CO2 Subcutaneous, right flank 2M<br>YUMM3.3 ATCC CRL-3365  37ºC 5% CO2 Subcutaneous, right flank 2M<br>YUMM5.2 ATCC CRL-3367  37ºC 5% CO2 Subcutaneous, right flank 2M<br>KPC-F (FC1245) Eric Collisson<br>(UCSF)  37ºC 5% CO2 Pancreas (orthotopic injections described in Jiang et al., Gastroenterology 2022) 1K<br>KPCY-C2 (6694c2)  37ºC 5% CO2  500K<br>KPCY-C5 (7160c5)  37ºC 5% CO2  500K<br>ID8 Fisher<br>Scientific SCC145 DMEM, 4% FCS, 1X Penicillin-Streptomycin-Glutamine, 1X Insulin-Transferrin-Selenium (ThermoFisher) 37ºC 5% CO2 Subcutaneous, right flank 2M<br>RENCA ATCC CRL-2947 RPMI (Gibco 11875-093), 10% FCS, 1X Penicillin-Streptomycin-Glutamine 37ºC 5% CO2 Subcutaneous, right flank 500K<br>CT26 ATCC CRL-2638  37ºC 5% CO2 Subcutaneous, right flank 500K |
|---|---|
| Authentication | No authentication was done in this study. |
| Mycoplasma contamination | All cell lines tested negative for mycoplasma. |
| Commonly misidentified lines<br>(See ICLAC register) | This study did not use misidentified lines. |

# Animals and other research organisms

Policy information about studies involving animals; ARRIVE guidelines recommended for reporting animal research, and Sex and Gender in Research

| Laboratory animals | Besides MMT-PyMT mice, this study used 7-9 week old female C57BL6 from JAX, stock No. 000664. |
|---|---|
| Wild animals | No wild animals were used in this study. |
| Reporting on sex | Except when indicated, experiments were performed using female mice. No different outcomes were observed based on sex differences. |
| Field-collected samples | This study did not involve field-sample collections. |
| Ethics oversight | All procedures were approved by the Institutional Animal Care and User Committee (IACUC). |

Note that full information on the approval of the study protocol must also be provided in the manuscript.

# Plants

| Seed stocks | N/A |
|---|---|
| Novel plant genotypes | N/A |
| Authentication | N/A |

# Flow Cytometry

## Plots

Confirm that:

☒ The axis labels state the marker and fluorochrome used (e.g. CD4-FITC).

☒ The axis scales are clearly visible. Include numbers along axes only for bottom left plot of group (a 'group' is an analysis of identical markers).

☒ All plots are contour plots with outliers or pseudocolor plots.

☒ A numerical value for number of cells or percentage (with statistics) is provided.

## Methodology

**Sample preparation**

CyTOF samples preparation:
After harvest, tumors were placed in a 12-well plate and minced to sub-millimeter pieces in 2mL of RPMI containing Collagenase IV (4 mg/mL) and DNase I (0.1mg/mL). Tissues were then incubated at 37℃ for 30 min, with a mechanical dissociation step using thorough pipetting after the first 15 minutes. Digestion was stopped by adding 2mL of cold RPMI + 10% FCS + 1X PS-Glu to each sample before filtering through a 100 mm mesh and centrifugation at 500 g for 5 min at 4℃. Cells were then resuspended in RPMI + 10% FCS + 1X PS-Glu for further counting and analysis.

Mass cytometry (CyTOF) was performed as described elsewhere76. Briefly, conjugations of mass cytometry antibodies with metal isotopes were done using the Maxpar® conjugation kit (Fluidigm) according to manufacturer's protocols and each antibody was titrated to define its optimal staining concentration. Each freshly digested sample was first stained with cisplatinium, fixed in 3.2% PFA and frozen at -80C. For CyTOF staining, the samples were then thawed and barcoded by mass-tag labelling with distinct combinations of stable Pd isotopes in 0.02% saponin in PBS before further pooling and staining. For this, cells were first resuspended in cell-staining media (Fluidigm) containing metal-labeled antibodies against CD16/32 for 5 min at room temperature to block Fc receptors, followed by the addition of a cocktail containing surface markers antibodies in a final volume of 500µL for 30 min at room temperature. Cells were then permeabilized with methanol for 10 min at 4 °C, washed and incubated with a cocktail containing intracellular markers antibodies in a final volume of 500µL for 30 min at room temperature (all antibodies listed in Table S2). Cells were finally stained with 191/193Ir DNA intercalator (Fluidigm) diluted in PBS with 1.6% PFA 48h prior to data acquisition. For acquisition, cells were washed and resuspended at 1M/mL in deionized water + 10% EQ four element calibration beads (Fluidigm) and analyzed on a CyTOF mass cytometer (Fluidigm).

scRNAseq samples preparation:
For most mouse experiment, we started by sampling 1e6 cells from the tumor of each animal and generated a single pool for each group. A group of 5 mice therefore generated a pool of 5e6 cells. This cell pool was then stained with the Zombie NIR viability dye (1/1000 in PBS, 10min at 4C), before being incubated with Fc block (clone 2.4G2, Tonbo Biosciences) and barcoded with HTO antibodies (TotalSeq-A from BioLegend). We then pooled all barcoded samples together and stained them with mix of fluorescent-labelled antibodies (Table S2). Using a BD FACSAria II cell sorter (BD Biosciences), we then gated live immune cells (Zombie-CD45+) and sorted 2 pools of cells from these samples: a pool of lymphoid cells containing equal amounts of T cells (CD90.2+), B cells (B220+MHCII+) and NK cells (CD49b+), and another pool of myeloid cells gated as CD11b + and/or CD11c+ among the non-T-B-NK cells. These two pools were then washed, counted and then individually encapsulated following 10X Genomics specifications for v.3 3' chemistry.

**Instrument**

BD LSR Fortessa X20 (BD Biosciences), LSRFortessa (BD Biosciences), CyTOF 2 mass cytometer (Fluidigm)

**Software**

BD FACSDiva, FlowJo

**Cell population abundance**

Abundances of relevant cell populations are described across the manuscript.

**Gating strategy**

Gating strategy is provided in supplementary figures.

☒ Tick this box to confirm that a figure exemplifying the gating strategy is provided in the Supplementary Information.

