## [Peer Review File · Nature Immunology]

Differential Assembly of Mouse and Human Tumor Microenvironments

Corresponding Author: Dr Matthew Krummel

Decision Letter:

10th Sep 2025

Dear Dr Krummel,

Your Resource, "Differential Assembly of Mouse and Human Tumor Microenvironments" has now been seen by 2 referees. You will see from their comments copied below that while they find your work of considerable potential interest, they have raised quite substantial concerns that must be addressed. In light of these comments, we cannot accept the manuscript for publication, but would be very interested in considering a revised version that addresses these serious concerns.

We hope you will find the referees' comments useful as you decide how to proceed. If you wish to submit a substantially revised manuscript, please bear in mind that we will be reluctant to approach the referees again in the absence of major revisions.

Please ensure to address all of the reviewers concerns, in particular concerns of R#1 regarding additional validation of datasets.

If you choose to revise your manuscript taking into account all reviewer and editor comments, please highlight all changes in the manuscript text file [OPTIONAL: in Microsoft Word format].

* If you have not done so already please begin to revise your manuscript so that it conforms to our Resource format instructions at <http://www.nature.com/ni/authors/index.html>. Refer also to any guidelines provided in this letter.

The Reporting Summary can be found here:

When submitting the revised version of your manuscript, please pay close attention to our <https://www.nature.com/nature-portfolio/editorial-policies/image-integrity> Digital Image Integrity Guidelines. and to the following points below:

Extended Data figures and tables are online-only (appearing in the online PDF and full-text HTML version of the paper),

peer-reviewed display items that provide essential background to the Article but are not included in the printed version of the paper due to space constraints or being of interest only to a few specialists. A maximum of ten Extended Data display items (figures and tables) is typically permitted. When re-submitting your manuscript, please ensure that any supplementary figures and tables that are more critical to the manuscript's conclusions are converted to Extended data to increase these data's visibility.

Link Redacted

If you wish to submit a suitably revised manuscript we would hope to receive it within 3 months. If you cannot send it within this time, please let us know. We will be happy to consider your revision so long as nothing similar has been accepted for publication at Nature Immunology or published elsewhere.

Nature Immunology is committed to improving transparency in authorship. As part of our efforts in this direction, we are now requesting that all authors identified as 'corresponding author' on published papers create and link their Open Researcher and Contributor Identifier (ORCID) with their account on the Manuscript Tracking System (MTS), prior to acceptance. ORCID helps the scientific community achieve unambiguous attribution of all scholarly contributions. You can create and link your ORCID from the home page of the MTS by clicking on 'Modify my Springer Nature account'. For more information please visit please visit www.springernature.com/orcid.

Thank you for the opportunity to review your work.

Sincerely,

Stephanie Houston, PhD
Senior Editor
Nature Immunology

Version 0:

Reviewer comments:

Reviewer #1

(Remarks to the Author)

The manuscript by Courau et al. analyzed 15 commonly used syngeneic murine tumor models (across both C57BL/6 and BALB/C backgrounds) to find key similarities and differences between the murine TME (based on new CyTOF and single-cell datasets generated in the current study) and the human TME (based on prior analyses from the group of human tumor samples). The resulting study provides a foundation for determining which models are most useful for mimicking specific human TME archetypes, and the data resource in this study will be helpful for the community to better understand the mouse models.

Several issues still need to be addressed to make sure the findings of the mouse to human comparisons are not confounded by technical issues. First, since composition depends on dissociation, which is quite variable in both the viabilities and isolation efficiencies of different cell types, additional validation of the claims are needed to make sure compositional differences are not caused by dissociation variability. The most definitive approach would be imaging of tissues (as they did to some extent in Combes et al. Cell 2022). Second, because different technologies were used in human and mouse studies for cell composition and gene expression measurements, the authors should consider how this could affect the findings, especially for cell types that may be hard to compare such as stromal cells (e.g. the stromal cells sorted from human tumors do not have a straightforward mouse equivalent). One feasible solution is to use external published single-cell RNAseq datasets to validate and deepen some of the results.

Comments

1. It is now always clear which samples, profiling methods and analysis approaches are used. The authors should: (1) provide a table of samples from mouse (and human) and which method was used to profile each sample; (2) clearer explanation in the main text of how many human samples profiled, how they were profiled and how they were used for each analysis shown; (3) provide more statistical tests for claims (e.g. for the 109 mouse samples in Fig 1, how many replicates

per tumor type, how were they combined to calculate the findings in each figure that uses them? See below for more examples).

2. The authors should say more about high-resolution subtypes within each cell type (e.g. stromal subtypes are never discussed but they are known to be important) and their composition per mouse tumor. For the mouse analysis (not human), the expression of chemokine/chemokine receptors could be shown for the subtypes within each lineage based on single-cell RNAseq, which would help readers understand the findings in this figure at higher resolution. The same point should be considered for GEP analysis in Fig. 4-6. This will be highly useful for investigators to decide which mouse model to use for different biological questions.

3. The authors' conclusion that murine TMEs are biased towards myeloid cells is a key finding that is well-supported by their data. However, this observation is not novel, as it was previously reported by Allen et al. (PMID: 32451499) and their results should be compared to that and other studies in the discussion.

4. There could be brief discussion of responsiveness of known models to checkpoint blockade based on the literature (assuming there is some consensus across studies), which could be helpful for integrating the knowledge of the field for readers.

5. More information is needed to understand how cell frequency determinations were made, and whether further validation is required. More specifically,

a. It's not clear why % live for CD45+ and for T cells is as high as 80-100% in some human tumors – does this mean tumor/epithelial cells were not viable after dissociation? If viability is highly variable, how does one determine true cell frequencies? And for mouse tumors, perhaps the T cell numbers are lower because there are more viable tumor cells? In addition, since stromal cells and macrophages are dissociated with lower efficiency compared to other cell types, their frequencies may not be accurate and could vary depending on the tissue and subtype of stromal cell or macrophage, this distorting the composition numbers.

b. While bulk RNAseq of tumors (Fig 2C) helps support the finding of immune deserts in mice, it would be important to know if the RNAseq measurements were made prior to any dissociation to avoid the issues above.

c. If the authors do not have a strong argument for why the current composition estimates are of high confidence (esp the high frequencies of CD45+ and T cells in humans in Fig 1B, C), the rigorous solution is to stain multiple sections (or cite publications) with multiplex antibody or FISH to validate the numbers found in dissociated samples by CyTOF/flow.

d. An important question is also how variable the composition is in the mouse tumors. How many replicates were used per mouse tumor type and how was the variability considered in the analysis? It would be helpful to show individual mice with cell numbers/frequencies in supplementary figures and tables, and then have stats to provide confidence intervals for the claims.

6. The relationships between mouse and human tumors is shown but not explained::

a. The reader may need more information to interpret the mapping of mouse to human tumor archetypes (Fig 2A/B). For example, Supp 1D shows the features of mouse tumors, but would be useful to also show human tumors (and their archetypes).

b. It's not clear exactly which are the mouse cell types/signatures that explain the mapping shown in 2C. It would be helpful to see a heatmap with key genes in the signature driving mouse tumors towards specific cancer archetypes (rather than just showing the % of tumors in each archetypes).

7. Fig 3 has a few questions to address:

a. Why do 3A and 3D not utilize the full scale from -2 to +2 (which should happen by definition)?

b. How can one justify the direct mouse-human comparisons for stroma given the mix of cell types in the sorted human cells vs. the more specific mouse subtypes by single cell analysis? Is there a way to match these better based on what's known of the human cell composition within the stromal sorting gate?

8. Fig 4 could be better explained and analyzed:

a. A bit more text describing the results would be useful along with limitations. For example, could the macrophage-CD8 correlation be missing in humans because of the sorting gate used for myeloid cells (HLA-DR+ which may not capture all myeloid subtypes)?

b. The 4D inset was quite useful by showing only desert archetypes. Can the same be displayed for 4B/C and even A? Perhaps even remove mouse RENCA for this comparison so that only deserts are shown.

9. There are many studies identifying the subtypes of myeloid and T cells in human tumors (e.g. papers from Zhemin Zhang and many others). The NMF analysis in the human should capture some of these, and test if their equivalent subtypes exist in the mice. Could the authors supplement their current approach (in Fig 5) with a direct comparison of single-cell-derived subtypes from publications to the mouse subtypes in this study? This would help overcome some of the limitations of the different technologies used across species, and make the resource more useful to the community. Perhaps this could be done with existing subtypes (e.g. similar to what was done in Fig 5H/I for GEPs similarity to those from the Gavish paper).

10. Can the authors develop statistical tests to determine which GEPS overlap between mouse and human with statistical significance?

Reviewer #2

(Remarks to the Author)

The authors performed a systematic comparison of tumor microenvironment compositions (TME) across species (murine and human) to describe fundamental similarities and differences by collecting high dimensional data (holistic views (e.g. archetypes or immune phenotypic features by CyTOF) and gene-expression programs (sc RNA seq for GEP). In 15 murine tumor models, they used either heterotopic cell lines (such as B16F10, MC38, CT26, LLC, and 4T1), autochthonous/transplanted (KPC) cell lines as well as genetically engineered (MMTV-PyMT) tumor models, all syngeneic of BALB/c or C57BL/6 mice, established in vivo for 14 days. The authors used their own human TME archetype registry (Combes et al. Cell. 2022 January 06; 185(1): 184–203.e19) to compare the murine TMEs presented in this current manuscript. They benchmarked their murine data with RNAseq data from a cohort of 2,846 mouse tumor samples (NCBI's Sequence Read Archive) and their human TME data with the TCGA human tumor sample data base. They concluded the following:

- 1/ Both species have a dichotomic distribution of total immune density ('rich' versus 'poor' immune infiltrates in murine & human tumors), but the global frequencies of immune cells in murine TME was significantly lower than that in human TME
- 2/ Focusing on 10 cell subtypes, they observed that murine TMEs contain lower frequencies of total T cells and higher frequencies of myeloid cells, i.e macrophages, compared to human TME; the variance in murine TMEs is mostly due to myeloid cells density and composition.
- 3/ all mouse models, except RENCA, grouped together with immune-desert, macrophage-rich, immune desert human samples. Indeed, RENCA clustered into immune-stromal CD4-biased (IS CD4) and the two myeloid centric archetypes (MC DC1 and MC DC2). They corroborated these findings using the NCBI and the TCGA data.
- 4/ The murine TME phenotype "immune desert, macrophage enriched" was also seen in orthotopic models, in dirty animal facilities, in aged mice and in high fat diet fed mice.
- 5/ CXCR3 and CXCR5 expression patterns were highly dissimilar among species. In murine TME, CXCL9/CXCL10 and CXCL13 transcripts were significantly enriched in the stroma while heavily expressed in myeloid and T cells/Treg areas respectively in human TME
- 6/ Intercellular networks highlight the positive correlation between CD8 T cell infiltrates and T cell exhaustion, and between CD8 and cDC1 & cDC2 but the anticorrelation between tumor Ki67 and CD8 density
- 7/ "T cell cytotoxicity" and "IFN response" GEP, that were correlated in the IR CD8 Mac, IS CD8 and ID CD8 Mac human archetypes and MC38, CT26 and B16-F10 mouse models were clinically relevant (associated with better survival in patients)

Overall, this is a very comprehensive and rigorous study that complements the first endeavor (Combes et al. in humans). Our community of tumor immunologist needs this type of repertoire and this array of resources. I have no further request.

Decision Letter:

Our ref: NI-RS40991A

13th Feb 2026

Dear Dr. Krummel,

Thank you for submitting your revised manuscript "Differential Assembly of Mouse and Human Tumor Microenvironments" (NI-RS40991A). It has now been seen by one of the original referees and an arbitrating referee, their comments are below. The original reviewer finds that the paper has improved in revision and the arbitrating reviewer is supportive of publication, therefore we'll be happy in principle to publish it in Nature Immunology, pending minor revisions to satisfy the referees' final requests and to comply with our editorial and formatting guidelines.

We will now perform detailed checks on your paper and will send you a checklist detailing our editorial and formatting requirements in about a week. Please do not upload the final materials and make any revisions until you receive this additional information from us.

If you had not uploaded a Word file for the current version of the manuscript, we will need one before beginning the editing process; please email that to immunology@us.nature.com at your earliest convenience.

Thank you again for your interest in Nature Immunology Please do not hesitate to contact me if you have any questions.

Sincerely,

Stephanie Houston, PhD
Senior Editor
Nature Immunology

Version 1:

Reviewer comments:

Reviewer #1

(Remarks to the Author)

We appreciate the authors' response to each comment, and addressing most of the key questions we raised.

Major comments

It would be helpful if the authors added in the limitations section of discussion that: a) this is not a comprehensive study of cell types (and that NK, B, neutrophils, innate lymphoid cells, etc are missing); b) the correspondence between human cell types derived by flow cytometry vs. mouse cell types derived by mass cytometry (or scRNA-seq) is not a head-to-head comparison (e.g. what makes up the group of 'myeloid' cells in each species is likely not the same), and that this may limit the biological interpretation of mouse model correspondence to human tumors.

Given these limitations (and the absence of direct comparisons of mouse dataset to human single-cell RNAseq datasets, which we suggested in our first review: "One feasible solution is to use external published single-cell RNAseq datasets to validate and deepen some of the results"), the study remains at relatively low resolution compared to what's feasible to do at this time with existing datasets. Nevertheless the dataset will be useful for the community.

Minor comments

The authors have now provided some mouse and human tumor tissue staining in Fig 1E/F to validate the T:myeloid cell ratio. This indeed provides more confidence in the estimates provided from flow for this strong difference between human and mouse. Minor point - please show markers used in legend for Fig 1E.

The authors addressed the variability in cell frequencies across individual mice in Figure S1B, and found that the variability was not too high. Please explain in the legend what is being shown in S1B since the reader cannot guess what the points are. However the ID8 tumor is highly variable for T and myeloid compartments, and in the UMAP. The authors should comment on this result since otherwise most readers would not find S1B and not appreciate ID8 variability, which could be technical or biological.

To show how mouse tumors were classified into human tumor archetypes by bulk RNA-seq, the authors provide a new Fig S1H showing consistency in the classification in mouse vs human for C1-C6 based on the signatures from Thorsson et al. However, I could not find a table with the classification of the 2,543 mouse tumors. The samples and assigned archetypes should be added as a tab to the current archetype-sample table.

The violin plot in Fig 1B indicates that some patients have more than 100% CD45⁺ cells relative to live cells. How did this happen? Will the raw data be available to readers to re-analyze the mass cytometry/flow cytometry data given that this is a useful resource to the community?

Reviewer #3

(Remarks to the Author)

The paper by Courau et al. takes advantage of new and previously published data to cross-compare the immune and non-immune stroma of cancer across humans and mice. This undertaking is of significant value to the field and, hopefully, to pharma, and it highlights the strengths and weaknesses of mouse models as a whole. I can clearly understand the importance of this work and of the dataset as a whole.

As I was not part of the first review, I have carefully reviewed the paper and the prior reviews. My thoughts align with Reviewer 1's initial points. This is a very strong submission to begin with. Also, in my view, the responses to each of the detailed reviews seem adequate.

My only additional point is that the community would be well served if the authors deposited some form of the analyzed data so that others could explore details of interest without having to rebuild the datasets. Where data are not shareable (e.g., human data), a tool or portal would be of significant utility. If such a resource already exists, better referencing it in the paper would likely draw in more readers..

Response to Reviewers

We thank both reviewers for their formative comments and particularly appreciate where both reviewers value the need for this systemic analysis of mouse tumor models, including in-depth analysis of which components of mouse models do, or do not, map to the spectrum of human tumor immune archetypes. We have addressed all requests with both text and experimental data/updates. We hope the work will now be ready for public consumption (thank you reviewer #2 for your support in moving this data into the public).

Reviewer #1 (Remarks to the Author in black, our replies in dark blue):

The manuscript by Courau et al. analyzed 15 commonly used syngeneic murine tumor models (across both C57BL/6 and BALB/C backgrounds) to find key similarities and differences between the murine TME (based on new CyTOF and single-cell datasets generated in the current study) and the human TME (based on prior analyses from the group of human tumor samples). The resulting study provides a foundation for determining which models are most useful for mimicking specific human TME archetypes, and the data resource in this study will be helpful for the community to better understand the mouse models.

We thank the reviewer for their support.

Several issues still need to be addressed to make sure the findings of the mouse to human comparisons are not confounded by technical issues. First, since composition depends on dissociation, which is quite variable in both the viabilities and isolation efficiencies of different cell types, additional validation of the claims are needed to make sure compositional differences are not caused by dissociation variability. The most definitive approach would be imaging of tissues (as they did to some extent in Combes et al. Cell 2022).

We fully agree that tissue dissociation can cause biases in both the efficiency of release and survival of cells during the process, which might lead to inaccurate tissue composition evaluation. We also agree that imaging of undissociated tissue is the ultimate approach to answer this question. Therefore, we have performed immunofluorescence on a variety of human and mouse tumors that we now include in our manuscript. Please see our specific replies, below.

Second, because different technologies were used in human and mouse studies for cell composition and gene expression measurements, the authors should consider how this could affect the findings, especially for cell types that may be hard to compare such as stromal cells (e.g. the stromal cells sorted from human tumors do not have a straightforward mouse

equivalent). One feasible solution is to use external published single-cell RNAseq datasets to validate and deepen some of the results.

We again appreciate the reviewer's point, here that using different technologies to compare mouse and human tumors, namely bulk vs single-cell RNAseq, can generate artificial differences between the two systems. We have included additional analyses and discussion to improve this point, again please see specific comments below. We believe our major findings are nicely both corroborated and extended by the additional analyses suggested. Thank you.

Comments

1. It is not always clear which samples, profiling methods and analysis approaches are used. The authors should: (1) provide a table of samples from mouse (and human) and which method was used to profile each sample; (2) clearer explanation in the main text of how many human samples profiled, how they were profiled and how they were used for each analysis shown; (3) provide more statistical tests for claims (e.g. for the 109 mouse samples in Fig 1, how many replicates per tumor type, how were they combined to calculate the findings in each figure that uses them? See below for more examples).

We agree this will help provide clarity and now provide with our manuscript a table presenting individual and summarized levels of description of each sample and analysis. You can find the table as **Table S1**.

2. The authors should say more about high-resolution subtypes within each cell type (e.g. stromal subtypes are never discussed but they are known to be important) and their composition per mouse tumor. For the mouse analysis (not human), the expression of chemokine/chemokine receptors could be shown for the subtypes within each lineage based on single-cell RNAseq, which would help readers understand the findings in this figure at higher resolution. The same point should be considered for GEP analysis in Fig. 4-6. This will be highly useful for investigators to decide which mouse model to use for different biological questions.

We thank the reviewer for pointing out that in the original submission we did not comment much on the granular resolution of cell subtypes in mouse tumors, even though **Figure S2** demonstrates this in UMAP and graphical form in substantial detail. We now added more analyses and description of those in the text at lines 221-230 to describe T cells (**S2B**), DC (**S2C**), Mono/Mac (**S2D**) and non-immune tumor and stroma (**S2E**) subtypes identity using UMAP and differential gene expression as well as frequencies across the 9 mouse models for which we have scRNAseq data. We specifically discuss the case of the Stroma compartment further in response to your comment 7b and will describe that more, below.

We also include in **Figure S3B-D** a detailed description of chemokine expression by each of the cell subtypes/subsets presented in **Figure S2**, which will hopefully help in understanding findings of **Figure 3** at higher resolution (mentioned in the manuscript at lines 240-243).

The same type of analysis was indeed warranted to explore GEP usage by each subtype, which is now described in **Figure S7C** (and here on the right) and commented in the text at lines 379-385. This analysis underlined varying patterns of GEPs usage across cell populations, including some GEPs that are specific to a given subtype (for example Mu_T_25 in gd T cells) while other GEPs seem to be used by different cell subsets (for example Mu_My_1 across Mono-DC, cDC2 and monocytes clusters).

3. The authors' conclusion that murine TMEs are biased towards myeloid cells is a key finding that is well-supported by their data. However, this observation is not novel, as it was previously reported by Allen et al. (PMID: 32451499) and their results should be compared to that and other studies in the discussion.

We agree that this observation has been lurking in the literature, reported specifically by Allen et al. although not quite so extensively as in this report and also not a primary focus of that work nor so not well-disseminated. We now added this reference to reinforce that this finding is at least supported by existing data in the literature, as well as a brief discussion of it at lines 494-495.

4. There could be brief discussion of responsiveness of known models to checkpoint blockade based on the literature (assuming there is some consensus across studies), which could be helpful for integrating the knowledge of the field for readers.

We now expand the existing discussion of this point which spans lines 555-570.

5. More information is needed to understand how cell frequency determinations were made, and whether further validation is required. More specifically,

a. It's not clear why % live for CD45+ and for T cells is as high as 80-100% in some human tumors – does this mean tumor/epithelial cells were not viable after dissociation? If viability is highly variable, how does one determine true cell frequencies? And for mouse tumors, perhaps the T cell numbers are lower because there are more viable tumor cells? In addition, since stromal cells and macrophages are dissociated with lower efficiency compared to other cell types, their

frequencies may not be accurate and could vary depending on the tissue and subtype of stromal cell or macrophage, this distorting the composition numbers.

We thank the reviewer for raising this point. This range of extremely high T cells frequencies in some human tumors was also very surprising to us at first. While it is possible (though difficult to demonstrate) that some tumor/epithelial cells might die more than T cells or other immune cells during dissociation, we think that these numbers may often reflect true large amounts of T cell infiltration in some human tumors. This is supported by existing literature, notably in kidney tumors (which is also where we observed this), that were also found to be often highly infiltrated by T cells (Mihecheva et al reported up to 83% of T cells in some patients). This also matches the Immunofluorescence data that we have generated on our ImmunoProfiler cohort. We agree that all frequencies that come from dissociation need to be taken with this grain of salt and one can only report what one gets from an assay as-described, but we sought at minimum to understand if frequencies of immune populations measured by flow scale with what is observed in imaging. Specifically to this point a. we show below (figure for reviewer only) that T cell frequencies calculated from flow cytometry and imaging on the same human samples correlate very well.

In addition, mouse and human samples were both dissociated for flow analyses, albeit the specifics of the tumor tissues vary across samples, but at least the same bias was applied to the 2 species. Nonetheless, with these kinds of correlation plots we feel confident that 'high' composition is at least a relative term that can be applied from flow cytometry. See our other remarks to your other subpoints, please.

b. While bulk RNAseq of tumors (Fig 2C) helps support the finding of immune deserts in mice, it would be important to know if the RNAseq measurements were made prior to any dissociation to avoid the issues above.

This is indeed a very important point, which brings major insight for this analysis as well as the general "dissociation-induced bias in TME composition" question. The data in question comes from a compendium of datasets and so we assessed the metadata of each mouse sample as reported in the literature. Because this cannot be done in an automated way at scale and thus must be examined manually, we instead performed a deep-dive into the top 20 studies

contributing the most samples (47% of the total samples, about 1340 samples) of **Figure 2C**. Overall, this led to the following updates and changes to the figure/text:

- We first found that one study (just 75 samples) used patient-derived xenograft samples grown in mice, that we decided now should be excluded from **Figure 2C** because they cannot be considered as “true murine tumors containing intact immune systems”. Interestingly, though, these samples contributed a majority of the poorly represented C6 “immune rich” subtype in mice. After updating our analysis (updated Figure 2C), ‘Immune rich’ tumors are now even more rarely found in mice as a percentage (4.7% → 2.1%).
- Two small studies (41 samples in total) were found using organoid and sorted cells, and we similarly excluded these from our revised analysis.
- After manually examining the Methods sections of the relevant papers and database metadata, we found that all the remaining studies were of non-dissociated samples.

Therefore, at least half of the samples in this massive analysis that reinforces the mapping of mouse models to immune deserts in human were analyzed without tissue dissociation.

We updated all graphs (existing **Figure 2C** and newly added **S1I**) and methods regarding this analysis (lines 745-757) to reflect changes made from this search, together with more stringent filters to exclude all PDX, organoid and sorted cells-originating samples. We thank the reviewer for this question, without which we might have included the hundred or so odd samples that slightly over-represent how similar mouse tumors are to human.

c. If the authors do not have a strong argument for why the current composition estimates are of high confidence (esp the high frequencies of CD45+ and T cells in humans in Fig 1B, C), the rigorous solution is to stain multiple sections (or cite publications) with multiplex antibody or FISH to validate the numbers found in dissociated samples by CyTOF/flow.

As mentioned above (point a.), we agree that visualizing and quantifying immune infiltration by imaging is a strong validation of the claims that we make using tissue dissociation-based frequencies. Therefore, we leveraged existing immunofluorescence data made on human FFPE tissue sections from the ImmunoProfiler cohort to extract the raw counts of T cells and myeloid cells across full-block sections of 85 patients and calculate their T:Myeloid cells ratio. In addition, we stained and generated new immunofluorescence data of a selection of mouse models (4xB16, 4xMC38 and 2xRENCA tumors, picked because of their range of overall immune and T cell infiltration as shown in **Figure S1B**) to calculate the same ratio and compare it cross-species. The result of this analysis is shown on the right and was added to our **Figure 1** and at lines 147-153. This demonstrates that most human tumors display a T:Myeloid ratio above one, while the 3 mouse models analyzed were very low for this ratio. This confirms the overall observation that human tumors are more typically dominated

for this ratio. This confirms the overall observation that human tumors are more typically dominated

by T cells and that conversely murine tumors are dominated by myeloid cells (as shown also by flow/CyTOF in **Figure 1C** and **S1F**).

d. An important question is also how variable the composition is in the mouse tumors. How many replicates were used per mouse tumor type and how was the variability considered in the analysis? It would be helpful to show individual mice with cell numbers/frequencies in supplementary figures and tables, and then have stats to provide confidence intervals for the claims.

We hope that our summary table in response to your comment 1 (**Table S1**) will help clarify this question. We also include in **Figure S1B** individual graphs showing the spread of each of the main compositional features (with means and standard deviations), across individual mouse samples grouped by tumor model to provide additional clarity.

6. The relationships between mouse and human tumors is shown but not explained:

a. The reader may need more information to interpret the mapping of mouse to human tumor archetypes (Fig 2A/B). For example, Supp 1D shows the features of mouse tumors, but would be useful to also show human tumors (and their archetypes).

We now include in **Figure 2A** (and here on the right) a version of **Figure S1D** that includes human samples so the readers can have a better understanding of the difference between human and murine samples when they are scaled together.

b. It's not clear exactly which are the mouse cell types/signatures that explain the mapping shown in 2C. It would be helpful to see a heatmap with key genes in the signature driving mouse tumors towards specific cancer archetypes (rather than just showing the % of tumors in each archetype).

The ImmuneSubtypeClassifier tool used to match mouse samples to human subtypes does not provide a straightforward scoring of genes per sample that we could use to address this point. However, we added more analyses to get around this issue and now include a heatmap in **Figure S1H** (and below) that answers this question.

This shows the output of scoring human TCGA vs mouse NCBI samples using the gene signatures used in Thorsson et al. The top genes contributing to each signature is annotated above each heatmap. It demonstrates that the pattern of murine-calculated signatures closely matches the pattern of human samples classified in the same subtype (added at lines 179-180 and legend of **S1H**). To provide additional details on this analysis, we also added a bar graph in **Figure S1I** that shows the contribution of each signature to each mouse tumor model or human immune subtype (see lines 185-187).

7. Fig 3 has a few questions to address:

a. Why do 3A and 3D not utilize the full scale from -2 to +2 (which should happen by definition)?

This is because the full scale from -2 to +2 is calculated from the complete set of genes and cell types shown in the heatmap in **Figure S3A/C**. The heatmaps in **Figure 3A/D** are subsets of these larger heatmaps. This was now added to the legend of this figure at line 937.

b. How can one justify the direct mouse-human comparisons for stroma given the mix of cell types in the sorted human cells vs. the more specific mouse subtypes by single cell analysis? Is there a way to match these better based on what's known of the human cell composition within the stromal sorting gate?

We agree that the similarity of human vs human stroma has not been discussed enough in our initial manuscript. Neither did we clearly describe what we meant by Stroma and how this compartment was isolated in this and our previous data collections. We now include new analysis in **Figure S2F** (and below) that describes our strategy on these points.

We used a human Stroma gene signature (that came from finding DGEs in the CD44+CD90+CD45- stromal sorted compartment and then showing that these strongly correlate with actual frequencies of CD44+CD90+ CD45- stromal populations in Combes et al. 2022. We note that this data was also mined in the work of Turley et al. and used as a measure of ‘myofibroblast’ content in their nomenclature). We then overlaid that on all our non-immune/tumor cells clusters from mouse samples and found that 3 of the clusters of mouse fibroblasts (that we here named Fibro. 1, Fibro. 2 and Myofibro. 1) harbored high expression of this signature. We thereafter considered these 3 clusters as being the mouse “Stroma” cells most equivalent to the human Stroma in Combes et al. 2022 and made comparisons of frequencies and gene expression based on this. These details now added to the manuscript at lines 800-803 and 1041-1049 as well as the **Figure S2F**.

8. Fig 4 could be better explained and analyzed:

a. A bit more text describing the results would be useful along with limitations. For example, could the macrophage-CD8 correlation be missing in humans because of the sorting gate used for myeloid cells (HLA-DR+ which may not capture all myeloid subtypes)?

We agree that this sorting strategy can be a confounding factor in this analysis. We discuss this point at lines 292-294.

b. The 4D inset was quite useful by showing only desert archetypes. Can the same be displayed for 4B/C and even A? Perhaps even remove mouse RENCA for this comparison so that only deserts are shown.

We agree that these insets make the reader appreciate better how these correlations become when only analyzing certain types of human tumor microenvironments. We now added similar insets for human graphs as well as RENCA-censored insets for mouse graphs in all panels of **Figure 4** and **Figure S4**.

9. There are many studies identifying the subtypes of myeloid and T cells in human tumors (e.g. papers from Zheming Zhang and many others). The NMF analysis in the human should capture some of these, and test if their equivalent subtypes exist in the mice. Could the authors supplement their current approach (in Fig 5) with a direct comparison of single-cell-derived subtypes from publications to the mouse subtypes in this study? This would help overcome some of the limitations of the different technologies used across species, and make the resource more useful to the community. Perhaps this could be done with existing subtypes (e.g. similar to what was done in Fig 5H/I for GEPs similarity to those from the Gavish paper).

This is a very interesting point. Previous studies attempted at finding consensus clusters to call similar subtypes of human vs murine myeloid cells (Zilionis et al.) and T cells (Andreatta et al.) in tumors and other tissues, and we think it is worth moving toward that in all future papers. However, as explained notably by Zilionis et al. it can be difficult to find clusters similarity simply based on DEGs because the top differentially expressed genes used by human vs murine cells often differ so strongly that they might be interpreted as signaling for different programs or subtypes. This is for example highlighted by the absence of CXCL13 or VCAM1 expression in murine CD8 T cells (that we show in **Figure 3** and **Figure 5C**) which are typically found in the top ~5-10 DEG of cytotoxic T cells in human. We think that GEP analysis may be better in future, as the question of 'what is a subtype' may prove to reduce to finding collections of GEPs.

The limitations posed by using different technologies in our study to compare human vs murine GEPs, that the reviewer rightfully pointed out, is yet extremely valid. So, since we indeed already used the Gavish study to compare human bulk vs scRNAseq-derived GEPs in T cells and myeloid cells (**Figure 5H/I**), we also now use it to compare **human vs mouse scRNAseq-calculated** GEPs factors in new **Figure S7B** (and below). This first confirms the conservation of factors described in **Figure 5B-G** (bolded colored fonts) but also identifies new similarities cross-species (boxed in

red) that we now expand on at lines 366-377. Most of these novel similarities were seen in T cells, where we find murine T_16, T_19, T_15 and T_23 to respectively map to GEPs named by Gavish et al. as proliferative, naïve, interferon-driven and glycolytic/dysfunctional human T cells.

The fact that we observe various degrees of GEPs overlap to specific clusters in mouse (in response to point 2 above, with new graph also added below for ease of readability) as well as significant mapping of human vs murine scRNAseq-generated GEPs in T cells and myeloid cells (our response above and graph below) tends to prove the reviewer right that some GEPs have correspondence to cell types in many atlases, and this type of analysis helps to identify new similarities cross-species as we describe. We believe our dataset and report, though not fully focusing on this aim, will substantially contribute to this type of analysis in future and the data will be available for further analyses of these issues by us and others, in those specific topic arenas.

10. Can the authors develop statistical tests to determine which GEPs overlap between mouse and human with statistical significance?

We thank the reviewer for this helpful suggestion and now added the results of Fisher's exact tests (comparing the odds of 2 list of genes to be significantly similar as compared to random comparisons across the constellation of all genes considered in the analysis) to all our Jaccard similarity analyses in **Figure 5, S6** and **S7**. These tests are a helpful addition as they now statistically validate our claims of similarity between different GEPs in these figures.

Reviewer #2 (Remarks to the Author in black, our replies in dark blue):

The authors performed a systematic comparison of tumor microenvironment compositions (TME) across species (murine and human) to describe fundamental similarities and differences by collecting high dimensional data (holistic views (e.g. archetypes or immune phenotypic features by CyTOF) and gene-expression programs (sc RNA seq for GEP). In 15 murine tumor models, they used either heterotopic cell lines (such as B16F10, MC38, CT26, LLC, and 4T1), autochthonous/transplanted (KPC) cell lines as well as genetically engineered (MMTV-PyMT) tumor models, all syngeneic of BALB/c or C57BL/6 mice, established in vivo for 14 days. The authors used their own human TME archetype registry (Combes et al. Cell. 2022 January 06; 185(1): 184–203.e19) to compare the murine TMEs presented in this current manuscript. They benchmarked their murine data with RNAseq data from a cohort of 2,846 mouse tumor samples (NCBI's Sequence Read Archive) and their human TME data with the TCGA human tumor sample data base. They concluded the following:

1/ Both species have a dichotomic distribution of total immune density

(‘rich’ versus ‘poor’ immune infiltrates in murine & human tumors), but the global frequencies of immune cells in murine TME was significantly lower than that in human TME

2/ Focusing on 10 cell subtypes, they observed that murine TMEs contain lower frequencies of total T cells and higher frequencies of myeloid cells, i.e macrophages, compared to human TME; the variance in murine TMEs is mostly due to myeloid cells density and composition.

3/ all mouse models, except RENCA, grouped together with immune-desert, macrophage-rich, immune desert human samples. Indeed, RENCA clustered into immune-stromal CD4-biased (IS CD4) and the two myeloid centric archetypes (MC DC1 and MC DC2). They corroborated these findings using the NCBI and the TCGA data.

4/ The murine TME phenotype ‘immuen desert, macrophage enriched” was also seen in orthotopic models, in dirty animal facilities, in aged mice and in high fat diet fed mice.

5/ CXCR3 and CXCR5 expression patterns were highly dissimilar among species. In murine TME, CXCL9/CXCL10 and CXCL13 transcripts were significantly enriched in the stroma while heavily expressed in myeloid and T cells/Treg areas respectively in human TME

6/ Intercellular networks highlight the positive correlation between CD8 T cell infiltrates and T cell exhaustion, and between CD8 and cDC1 &cDC2 but the anticorrelation between tumor Ki67 and CD8 density

7/‘T cell cytotoxicity’ and “IFN response’ GEP, that were correlated in the IR CD8 Mac, IS CD8 and ID CD8 Mac human archetypes and MC38, CT26 and B16-F10 mouse models were clinically relevant (associated with better survival in patients)

Overall, this is a very comprehensive and rigorous study that complements the first endeavor (Combes et al. in humans). Our community of tumor immunologist needs this type of repertoire and this array of resources. I have no further request.

We warmly thank the reviewer for this report. We are glad to have met this important need for the cancer immunology community.